# Amotl1 mediates sequestration of the Hippo effector Yap1 downstream of Fat4 to restrict heart growth

Chiara V. Ragni[1,2,3,*], Nicolas Diguet[1,2,*], Jean-François Le Garrec[1,2,†], Marta Novotova[4], Tatiana P. Resende[5,6], Sorin Pop[7,8], Nicolas Charon[9,10], Laurent Guillemot[1,†], Lisa Kitasato[†], Caroline Badouel[11], Alexandre Dufour[7,8], Jean-Christophe Olivo-Marin[7,8], Alain Trouvé[9,10], Helen McNeill[11] & Sigolène M. Meilhac[1,2,†]

Although in flies the atypical cadherin Fat is an upstream regulator of Hippo signalling, the closest mammalian homologue, Fat4, has been shown to regulate tissue polarity rather than growth. Here we show in the mouse heart that Fat4 modulates Hippo signalling to restrict growth. *Fat4* mutant myocardium is thicker, with increased cardiomyocyte size and proliferation, and this is mediated by an upregulation of the transcriptional activity of Yap1, an effector of the Hippo pathway. Fat4 is not required for the canonical activation of Hippo kinases but it sequesters a partner of Yap1, Amotl1, out of the nucleus. The nuclear translocation of Amotl1 is accompanied by Yap1 to promote cardiomyocyte proliferation. We, therefore, identify Amotl1, which is not present in flies, as a mammalian intermediate for non-canonical Hippo signalling, downstream of Fat4. This work uncovers a mechanism for the restriction of heart growth at birth, a process which impedes the regenerative potential of the mammalian heart.

[1] Institut Pasteur, Department of Developmental and Stem Cell Biology, 75015 Paris, France. [2] CNRS URA2578, 75015 Paris, France. [3] Sorbonne Universités, UPMC Université Paris 06, IFD, 4 Place Jussieu, 75005 Paris, France. [4] Institute of Molecular Physiology and Genetics, Centre of Biosciences, Slovak Academy of Sciences, Dúbravská cesta 9, 84005 Bratislava, Slovak Republic. [5] Instituto de Investigação e Inovação em Saúde (i3S), Universidade do Porto, 4200-135 Porto, Portugal. [6] Instituto de Engenharia Biomédica (INEB), Universidade do Porto, 4200-135 Porto, Portugal. [7] Institut Pasteur, Quantitative Image Analysis Unit, 75015 Paris, France. [8] CNRS URA 2582, 75015 Paris, France. [9] ENS Cachan, Center of Mathematics and Their Applications, 94235 Cachan, France. [10] CNRS UMR 8536, 94235 Cachan, France. [11] Samuel Lunenfeld Research Institute, Mt Sinai Hospital, Toronto, Ontario, Canada M5G 1X5. * These authors contributed equally to this work. † Present address: *Imagine*-Institut Pasteur, Laboratory of Heart Morphogenesis, INSERM UMR1163, 75015 Paris, France. Correspondence and requests for materials should be addressed to S.M.M. (email: sigolene.meilhac@inserm.fr).

The growth of the mammalian heart is critical for its contractile function. During development, cell proliferation underlies most of the growth, whereas increase in cell size (hypertrophy) predominates after birth[1]. The growth of the mammalian heart is controlled by the Hippo pathway[2–5]. Hippo kinases[3] and Hippo effectors[4,5] are required to regulate heart growth during development, and can also be manipulated to re-activate cardiomyocyte division in the postnatal heart, thus improving heart repair after injury[6,7]. However, upstream regulators of the Hippo pathway in this context have remained unknown.

In flies, the atypical cadherin, Fat, is an upstream regulator of Hippo signalling[8], functioning as a tumour suppressor, and is also a regulator of tissue polarity[9,10]. Fat controls the canonical Hippo pathway by activating the Hippo kinases, which phosphorylate the Yap1 homologue, Yorkie, to sequester it in the cytoplasm, preventing transcriptional activation of target genes involved in cell proliferation and survival[11]. Another level of regulation of Yorkie downstream of Fat is provided by the FERM-domain protein Expanded, which requires Fat for its localization at the membrane[8] and thus sequesters Yorkie by direct binding[12]. Expanded is also localized at the membrane by the cell junction protein Crumbs[13].

In mammals, Hippo signalling has similarly been shown to be modulated by cell junction proteins, and this is mediated in the early embryo (blastocyst) by the adaptor protein Angiomotin, which interacts with Yap1 (ref. 14). In the liver, Angiomotin can translocate to the nucleus together with Yap1, thereby activating cell proliferation[15]. Mouse mutants for the closest homologue of *Drosophila* Fat, Fat4, die at birth with polarity defects, exemplified by the kidney or cochlea[16,17]. *Fat4* mutations are also associated with abnormal cortical development in Van Maldergem syndrome, due to the misregulation of Yap1 which affects cell differentiation[18]. However, in the cortex, kidney or liver of *Fat4* mutants, no activation of the Hippo kinases has been reported[12,16–19]. The mouse Fat4 intracellular domain cannot rescue the growth phenotype of *fat* mutant flies[20], raising the question of whether and how Fat4 could regulate the Hippo pathway in the mouse.

We show that Fat4 is required to restrict heart growth at birth, by decreasing the activity of Yap1. This regulation does not involve the phosphorylation cascade of Hippo kinases, but rather the scaffold protein Angiomotin-like 1 (Amotl1), a partner of Yap1 not present in flies. When it is not sequestered by Fat4, Amotl1 can translocate to the nucleus together with Yap1, thereby activating cardiomyocyte proliferation.

## Results

**Fat4 is required to restrict heart growth.** In the heart, we assessed the phenotype of $Fat4^{-/-}$ mutants at birth and observed an abnormal flattened apex (Fig. 1a) and 1.5-fold thicker ventricular myocardium and interventricular septum (Fig. 1b,c), compared with control hearts. This excessive tissue thickness may result from defective orientation or excessive rate of growth. We had shown previously that clonal growth of the heart is oriented[21,22], and we, therefore, analysed the orientation of cell division in $Fat4^{-/-}$ mutants. Whereas in the mouse kidney orientation of cell division is disrupted when Fat4 is absent[17], we found that cell division was still biased in the plane of the heart surface and still coordinated in the left ventricle of $Fat4^{-/-}$ hearts as in controls (Supplementary Fig. 1). We then investigated whether the excessive growth of Fat4 mutant hearts was due to increased cell proliferation, and found that the percentage of cardiomyocytes positive for the mitotic marker, phosphorylated histone H3, was significantly higher in the absence of Fat4 (Fig. 2a,b), whereas the effect was milder in non-cardiomyocytes

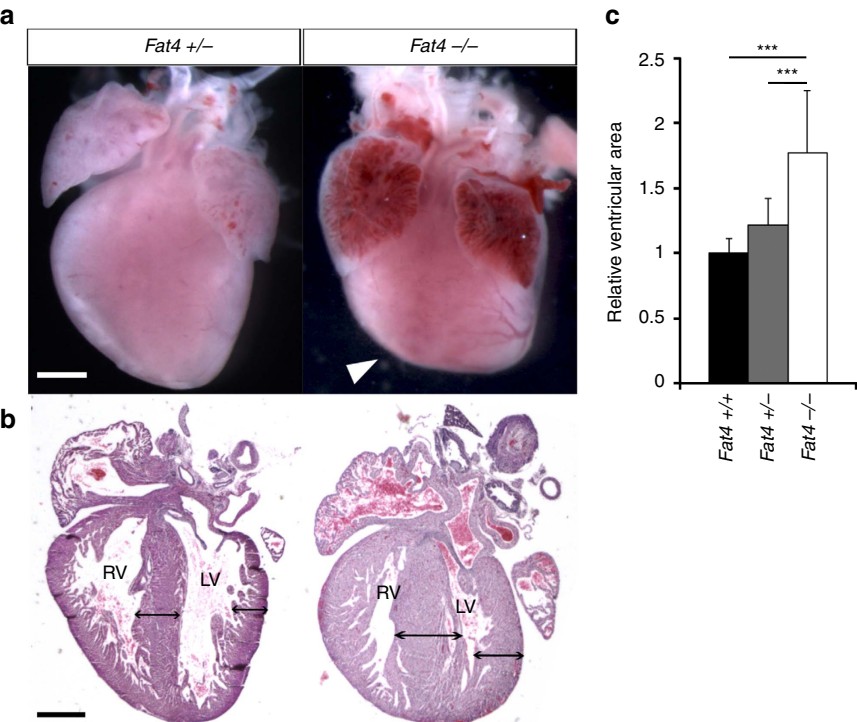

**Figure 1 | Excessive thickness of Fat4 mutant hearts.** Whole mount views (**a**) and histological sections (**b**) of $Fat4^{+/-}$ and $Fat4^{-/-}$ neonatal (P0) hearts. The arrowhead points to the flattened apex and double arrows highlight ventricular wall and septum thickness. (**c**) Quantification of the thickening in $Fat4^{+/+}$ ($n=7$), $Fat4^{+/-}$ ($n=5$) and $Fat4^{-/-}$ ($n=6$) hearts. ***$P<0.001$ (analysis of variance). Scale bar: 500 μm. RV, right ventricle; LV, left ventricle. In all figures, data are presented as means ± standard deviations, normalized to the level of control hearts when appropriate.

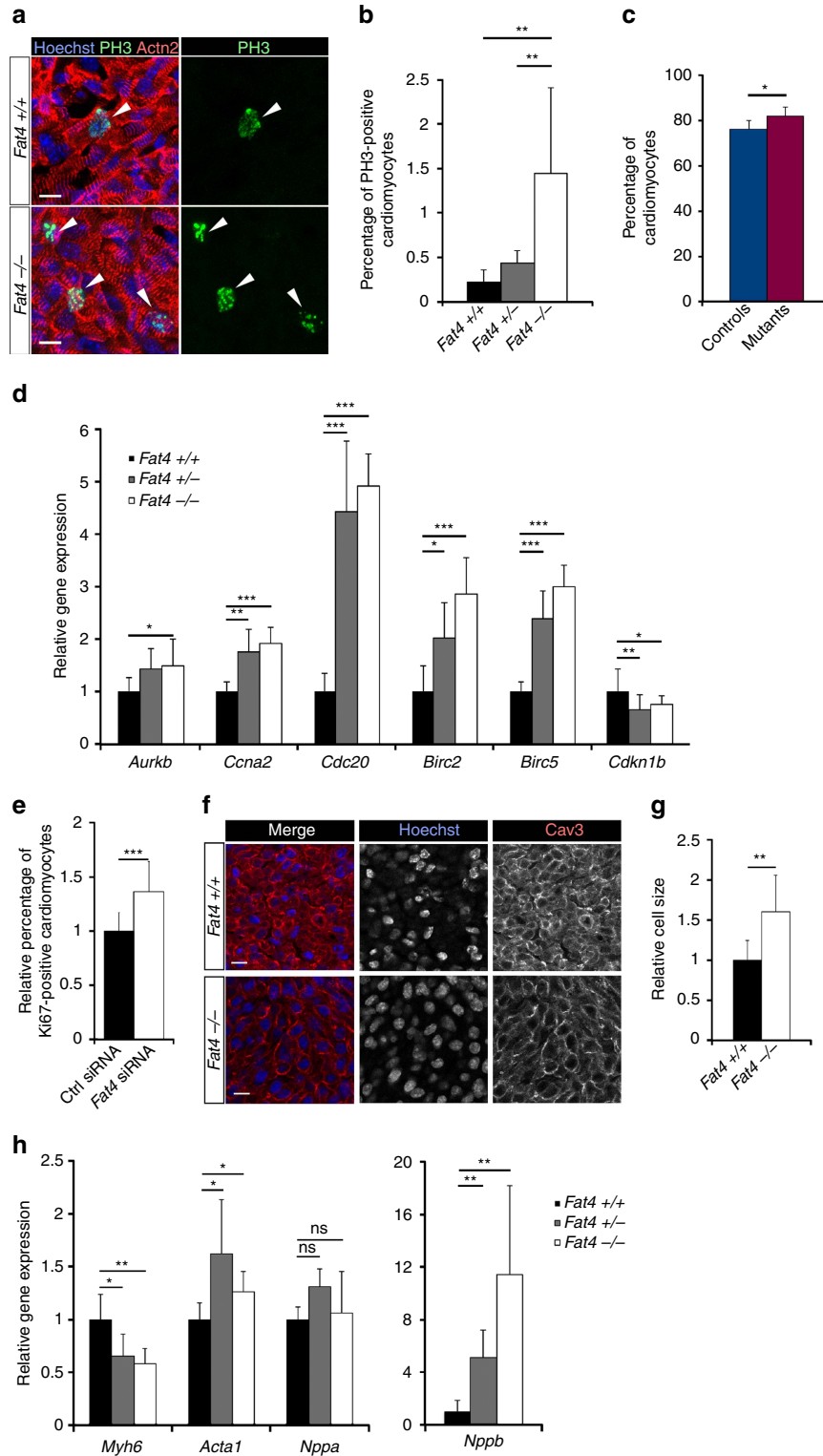

**Figure 2 | Fat4 restricts cell proliferation and hypertrophy.** Immunodetection (**a**) and quantification (**b**) of the percentage of mitotic cardiomyocytes positive for phosphorylated histone H3 (PH3) (arrowheads) in $Fat4^{+/+}$ ($n=6$), $Fat4^{+/-}$ ($n=6$) and $Fat4^{-/-}$ ($n=5$) hearts at P0. **$P<0.01$ (analysis of variance (ANOVA)). (**c**) Percentage of cardiomyocytes per total number of cells, analysed by ImageStream, in $Fat4^{Flox/+}$ (controls, $n=7$) and $Fat4^{Flox/-}$;$Mesp1^{Cre/+}$ (mutants, $n=6$) hearts at P14. *$P<0.05$ (Student test). (**d**) Relative transcript levels of cell cycle ($Aurkb$, $Ccna2$, $Cdc20$), cycle exit ($Cdkn1b$) or survival ($Birc2/5$) genes in $Fat4^{+/+}$ ($n=5$), $Fat4^{+/-}$ ($n=5$) or $Fat4^{-/-}$ ($n=7$) hearts at P0. *$P<0.05$, **$P<0.01$, ***$P<0.001$ (ANOVA). (**e**) Increased percentage of proliferating Ki67-positive cardiomyocytes, as counted by flow cytometry from primary cultures of neonatal rat cardiomyocytes treated with $Fat4$ siRNA ($n=20$) compared with control (ctrl) cultures ($n=20$). ***$P<0.001$ (Student test). (**f**) Immunodetection of cardiomyocyte cross-sectional area, with an antibody to Caveolin3. (**g**) Quantification of this in the interventricular septum indicates cell hypertrophy in $Fat4^{-/-}$ ($n=6$) compared with $Fat4^{+/+}$ ($n=9$) hearts at P0. **$P<0.01$ (Student test). (**h**) Expression of positive ($Nppb$ and $Acta1$ (skeletal actin)) and negative ($Myh6$ (α myosin heavy chain)) hypertrophy markers and of a marker of wall stress ($Nppa$) quantified by RT-qPCR in $Fat4^{+/+}$ ($n=5$), $Fat4^{+/-}$ ($n=5$) or $Fat4^{-/-}$ ($n=7$) hearts at P0. *$P<0.05$, **$P<0.01$ (ANOVA). ns, no significant difference. Scale bars: 10 µm.

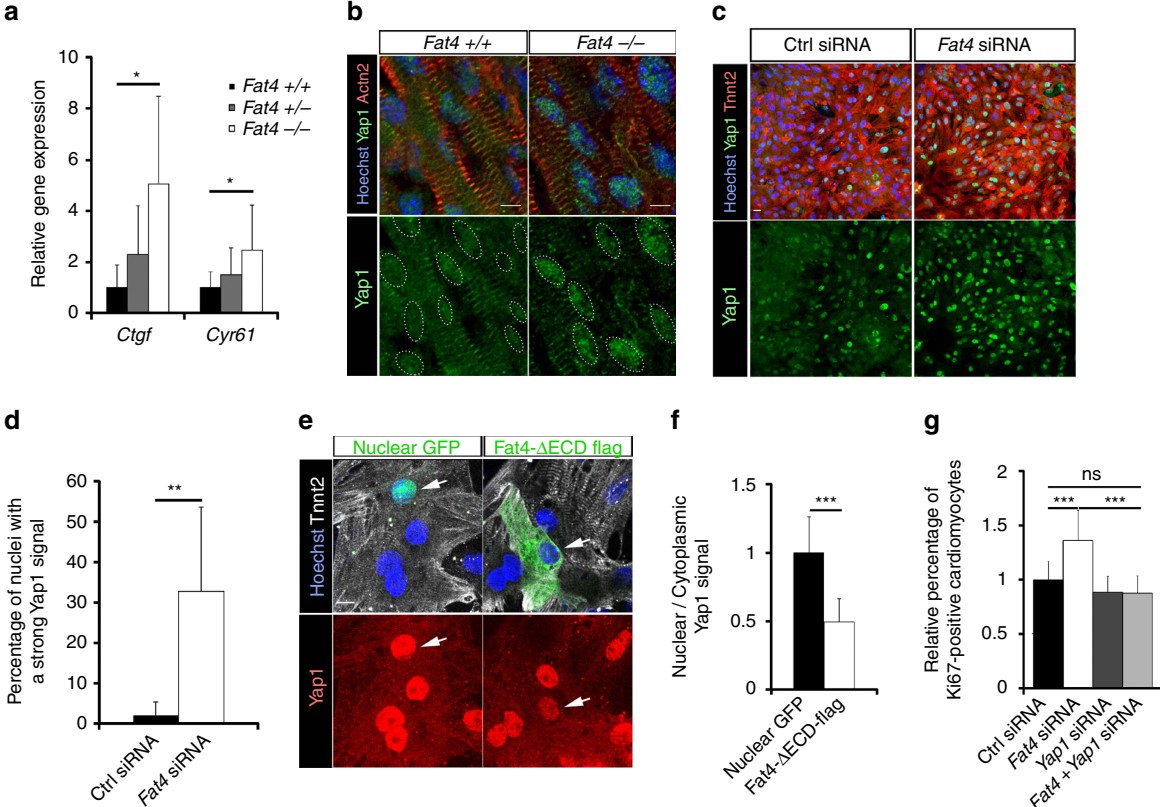

**Figure 3 | Fat4 modulates Hippo signalling.** (**a**) Relative expression of Yap1 target genes in $Fat4^{+/+}$ ($n = 5$), $Fat4^{+/-}$ ($n = 5$) or $Fat4^{-/-}$ ($n = 7$) hearts at P0. $*P < 0.05$ (analysis of variance (ANOVA)). (**b**) Immunodetection of Yap1 localization in the heart of the indicated genotype at P0. Nuclei are encircled with dashed white lines. (**c**) Immunodetection of Yap1 localization in primary cultures of neonatal rat cardiomyocytes treated with the indicated siRNA. (**d**) Quantification of the increased percentage of nuclei with a strong Yap1 signal when Fat4 is downregulated ($n = 6$ cultures) relative to the control ($n = 6$). $**P < 0.01$ (Student test). (**e**) Immunodetection of Yap1 (nuclear, arrows) localization in primary cultures of cardiomyocytes transfected with the indicated plasmids. Fat4-ΔECD-Flag, which is depleted for the extracellular domain (ECD), is used for overexpression. (**f**) Quantification of its nuclear to cytoplasmic localization, which is decreased when Fat4 is overexpressed ($n = 14$ cells) relative to control GFP ($n = 9$). $***P < 0.001$ (Student test). (**g**) The percentage of proliferating Ki67-positive cardiomyocytes, counted by flow cytometry from primary cell cultures, indicates a genetic rescue of Fat4 siRNA by Yap1 siRNA ($n = 20$ ctrl, 20 Fat4, 8 Yap1 and 8 Fat4 + Yap1 siRNA cultures). $***P < 0.001$ (ANOVA). ns, no significant difference. Scale bars: 5 μm in **b**, 20 μm in **c**, 10 μm elsewhere.

(Supplementary Fig. 2a). Mutant cardiomyocytes undergo cytokinesis, as revealed by Aurkb staining (Supplementary Fig. 2b). Conditional deletion of *Fat4*, in cardiac progenitors that express *Mesp1*, resulted in an increased percentage of cardiomyocytes, whereas the number of nuclei per cell was unchanged (Fig. 2c, Supplementary Fig. 2c). This supports the conclusion of increased cardiomyocyte proliferation. Conditional deletion of *Fat4* in non-cardiomyocytes, using *Wt1-Cre*, did not change the thickness of the ventricular wall or the percentage of mitotic cells (Supplementary Fig. 2d–f), indicating that Fat4 is required in cardiomyocytes. Analysis of gene expression showed an increase for cell cycle (*Ccna2*, *Cdc20*), cytokinesis (*Aurkb*) and anti-apoptotic (*Birc2/5*) genes and a decrease of *Cdkn1b* transcripts implicated in cell cycle exit, in $Fat4^{-/-}$ compared with control hearts (Fig. 2d). Heterozygotes also show transcript upregulation, although they do not have a detectable heart phenotype, indicating compensation at the level of the proliferation gene network dependent on Fat4 dosage. In primary cultures, knockdown of *Fat4* (Supplementary Fig. 2g–k) significantly enhanced the percentage of proliferating Ki67-positive and replicating EdU-positive cardiomyocytes (Fig. 2e and Supplementary Fig. 2l,m). We next examined whether Fat4 also affects cell size. By measuring the cross sectional area of cardiomyocytes, we found a significant increase of cell size in

$Fat4^{-/-}$ compared with control hearts (Fig. 2f,g). In agreement with a hypertrophic phenotype, reverse transcription quantitative PCR (RT–qPCR) revealed a de-regulation of classical markers of heart hypertrophy[23], corresponding to the activation, in $Fat4^{-/-}$ mutant hearts, of genes normally expressed at fetal stages (*Acta1*), whereas genes normally expressed at adult stages (*Myh6*) are downregulated (Fig. 2h). The early marker of heart hypertrophy, *Nppb*[24], was strikingly increased (11-fold) in $Fat4^{-/-}$ mutant hearts, whereas the marker of wall stress, *Nppa*, was not. Our data show that Fat4 is required to restrict heart growth at birth, with an effect on the proliferation and hypertrophy of cardiomyocytes.

**Fat4 modulates an effector of Hippo signalling Yap1.** Since we observed misexpression of genes (Fig. 2d) previously shown to be targets of the Hippo pathway in the control of cardiomyocyte proliferation[3,4], we examined whether Hippo signalling was impaired in *Fat4* mutant hearts. The classical targets of Yap1, *CTGF* and *Cyr61*, were significantly overexpressed (Fig. 3a), showing that the transcriptional activity of Hippo effectors is increased in $Fat4^{-/-}$ mutant hearts. Consistent with this, Yap1 was relocalized to the nucleus of cells in which Fat4 was absent *in vivo* (Fig. 3b) or downregulated in primary cultures of cardiomyocytes (Fig. 3c,d, Supplementary Fig. 3a,k,l), and

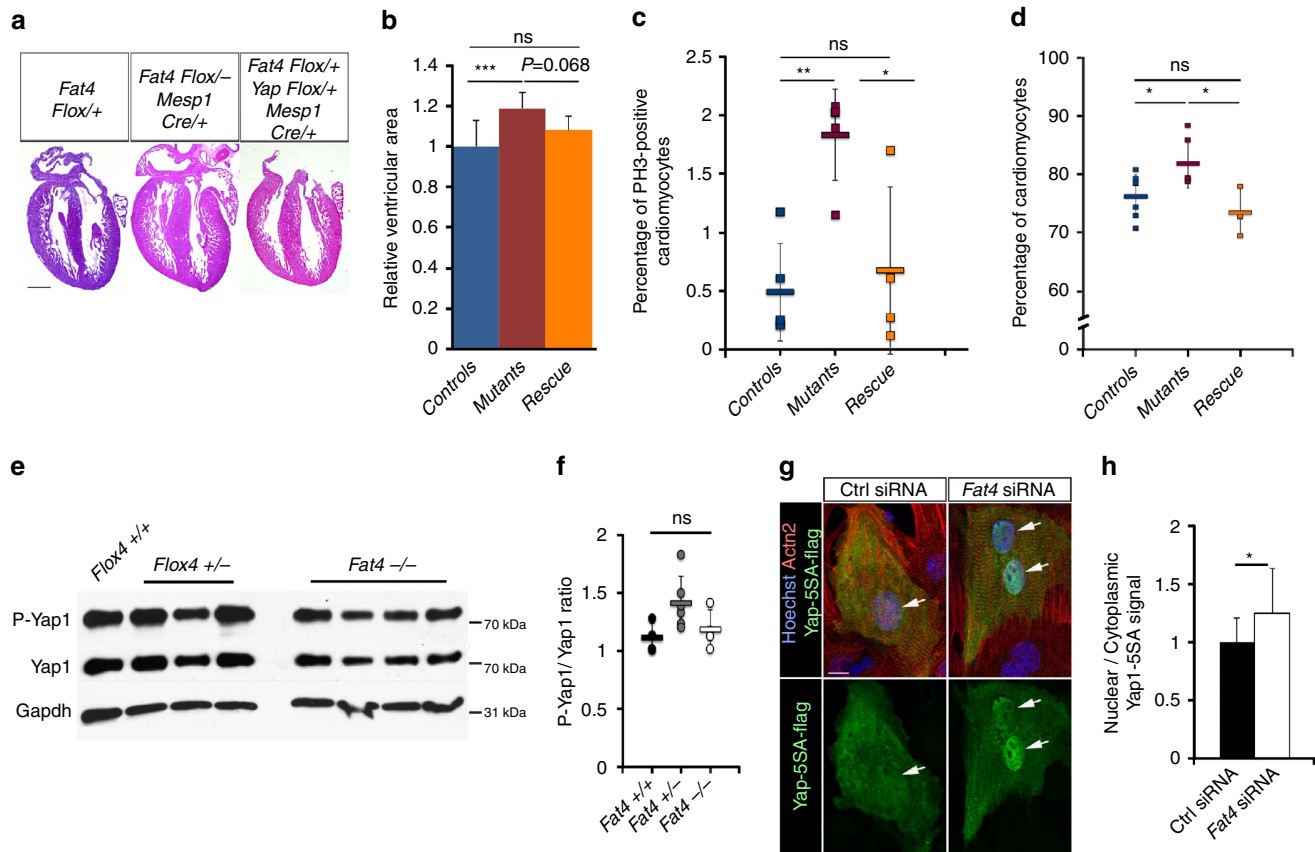

**Figure 4 | Fat4 modulates non-canonical Hippo signalling.** (**a**) Histological sections of control $Fat4^{Flox/+}$, conditional mutant $Fat4^{Flox/-};Mesp1^{Cre/+}$ and rescued $Fat4^{Flox/-};Yap1^{Flox/+};Mesp1^{Cre/+}$ hearts at P0. (**b**) Corresponding quantification of the ventricular thickening ($n=6$ ctrl, 6 mutant, 6 rescued hearts). ***$P<0.001$ (analysis of variance, (ANOVA)). (**c**) Quantification of the percentage of mitotic cardiomyocytes positive for PH3 in $Fat4^{Flox/+}$ (controls, $n=5$), $Fat4^{Flox/-};Mesp1^{Cre/+}$ (mutants, $n=5$) and $Fat4^{Flox/-};Yap1^{Flox/+};Mesp1^{Cre/+}$ (rescue, $n=4$) hearts at P0. *$P<0.05$, **$P<0.01$ (ANOVA). (**d**) Percentage of cardiomyocytes per total number of cells, analysed by ImageStream, in $Fat4^{Flox/+}$ (controls, $n=7$), $Fat4^{Flox/-};Mesp1^{Cre/+}$ (mutants, $n=6$) and $Fat4^{Flox/-};Yap1^{Flox/+};Mesp1^{Cre/+}$ (rescue, $n=3$) hearts at P14. *$P<0.05$ (ANOVA). (**e**) Western blot showing normal Yap1 phosphorylation at the Hippo kinase target site. (**f**) Corresponding quantification, from the blot shown in Supplementary Fig. 3b, in $Fat4^{+/+}$ ($n=4$), $Fat4^{+/-}$ ($n=6$) or $Fat4^{-/-}$ ($n=4$) hearts at P0. (**g**) Immunodetection of the localization of phospho-resistant Yap1 (Yap5SA) in primary cultures of cardiomyocytes treated with the indicated siRNA. (**h**) Quantification of its nuclear to cytoplasmic localization, which is increased when Fat4 is downregulated ($n=6$ cultures) relative to the control ($n=6$). *$P<0.05$ (Student test). Ctrl, control; ns, no significant difference. Scale bars: 500 μm in **a**, 10 μm elsewhere.

reduced in the nucleus of cells in which Fat4 was overexpressed (Fig. 3e,f). Knockdown of *Yap1* transcripts by RNA interference (Supplementary Fig. 2g–k) rescued the increased cell proliferation observed when Fat4 is downregulated (Fig. 3g). Deletion of one copy of *Yap1* similarly rescued the excessive number of mitotic cells in *Mesp1-Cre* conditional *Fat4* mutant hearts, as well as the percentage of cardiomyocytes, with a clear tendency to rescue the thickness of the ventricular wall (Fig. 4a–d). As *Mesp1-Cre* targets 70% of cardiac cells (Supplementary Fig. 2n), the conditional *Fat4* mutant hearts have a milder phenotype than the constitutive $Fat4^{-/-}$ hearts. The rescue experiments with a knockdown of *Yap1* are consistent with the proposition that Fat4 acts upstream of Yap1. To examine whether canonical Hippo signalling was activated, as in the fly model, we analysed the phosphorylation of Yap1 at the Ser[127] target site of Hippo kinases. Neither *in vivo*, nor in primary cell cultures (Fig. 4e,f, Supplementary Fig. 3b–f), could we detect a significant change in the ratio of phosphorylated Yap1, over total Yap1, when Fat4 was downregulated. The phosphorylation of the Hippo kinases Lats and Mst was also unaffected when Fat4 is absent (Supplementary Fig. 3g–j). A phospho-resistant form of Yap1 was relocalized to the nucleus when Fat4 was downregulated (Fig. 4g,h), further

supporting the conclusion that the nuclear translocation of Yap1 downstream of Fat4 is not mediated by a change of phosphorylation. The phenotype of *Fat4* mutants differs from that resulting from impairment of the canonical Hippo pathway. The onset of excessive myocardial growth in Yap1 gain-of-function mutants[5] or in Hippo kinase-deficient hearts[3] is already seen at embryonic stages (E10.5-E11.5). In contrast, the phenotype of $Fat4^{-/-}$ hearts is detected much later, from E18.5, although *Fat4* is expressed throughout heart development (Supplementary Fig. 4a–c). The Wnt pathway, which was previously shown to interact with the canonical Hippo pathway, was not found activated in $Fat4^{-/-}$ mutant hearts (Supplementary Fig. 4d,e). These observations show that Fat4 is a later modulator of the Hippo pathway and suggest that a mechanism other than that of phosphorylation by the Hippo kinases, regulates the nuclear localization of Yap1, downstream of Fat4.

**Amotl 1 mediates Fat4 signalling.** Hippo signalling is modulated by cell junction proteins[25]. When we labelled cardiomyocytes with junction markers, we observed that cell junctions

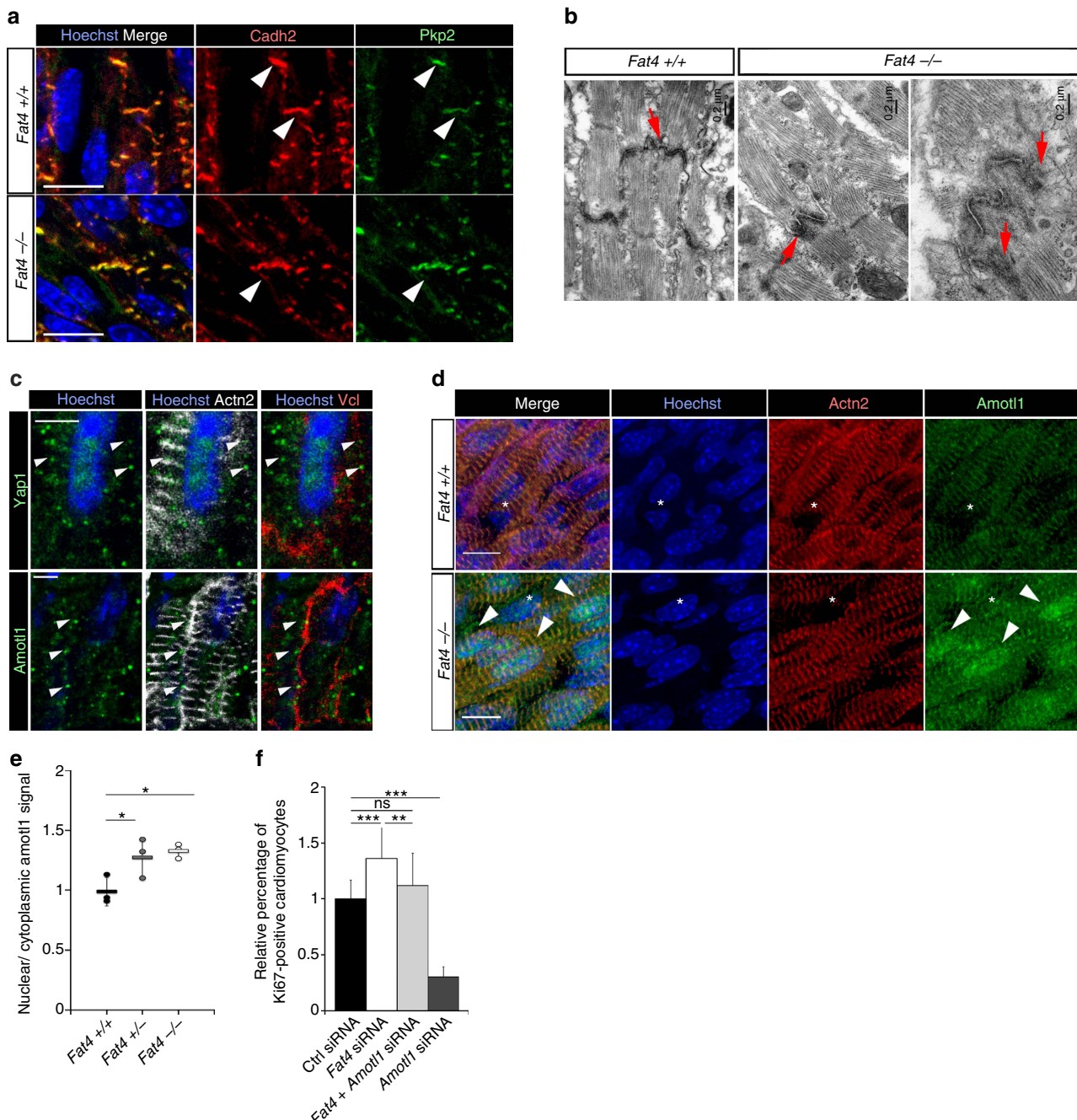

**Figure 5 | Fat4 signalling is required for the integrity of cell junctions and the localization of Amotl1. (a)** Immunodetection of cell junctions (arrowheads), marked by N-cadherin (Cadh2) and Plakophilin2 (Pkp2), in E18.5 hearts. They are disorganized in the absence of Fat4. **(b)** Transmission electron micrographs of $Fat4^{+/+}$ and $Fat4^{-/-}$ hearts at P0. The electron-dense material of desmosomes (arrows) in the intercalated discs is abnormally spread in mutant hearts. **(c)** Immunodetection of Amotl1 and Yap1 (white arrowheads) at similar positions near the cell membrane, marked by vinculin (Vcl), of cardiomyocytes, marked by α-actinin (Actn2) in P0 control hearts. **(d)** Amotl1 is relocalized to cardiomyocyte nuclei (arrowheads) in $Fat4^{-/-}$ hearts at E18.5. *, non-cardiomyocyte nuclei. **(e)** Quantification of nuclear to cytoplasmic localization of Amotl1 in $Fat4^{+/+}$ ($n = 3$), $Fat4^{+/-}$ ($n = 3$) or $Fat4^{-/-}$ ($n = 3$) hearts. *$P < 0.05$ (analysis of variance (ANOVA)). **(f)** The percentage of proliferating Ki67-positive cardiomyocytes, counted by flow cytometry from primary cultures of neonatal rat cardiomyocytes, indicates a genetic rescue of Fat4 siRNA by Amotl1 siRNA and lower proliferation with Amotl1 siRNA ($n = 20$ ctrl, 20 Fat4, 16 Amotl1 and 16 Fat4 + Amotl1 siRNA cultures). **$P < 0.01$, ***$P < 0.001$ (ANOVA). Ctrl, control; ns, no significant difference. Scale bars: 0.2 μm in **b**, 5 μm in **c**, 10 μm elsewhere.

were abnormal in $Fat4^{-/-}$ hearts. N-cadherin (Cadh2) or Plakophilin2 (Pkp2) staining were broader and less focalized than in control hearts (Fig. 5a). By electron microscopy, the electron dense desmosomal material was more diffuse in cardiomyocytes of $Fat4^{-/-}$ hearts (Fig. 5b). In agreement with abnormal cell junctions, cardiomyocytes had a rounder shape in *Mesp1-Cre*

conditional Fat4 mutant hearts (Supplementary Fig. 4f). These observations suggest that a junctional protein may be involved in the effect of the atypical cadherin Fat4 on Hippo effectors. We focussed on the adaptor protein, Angiomotin, which had been shown to interact with Yap1 and to co-translocate to the nucleus, where the complex modulates transcription[15]. This co-translocation

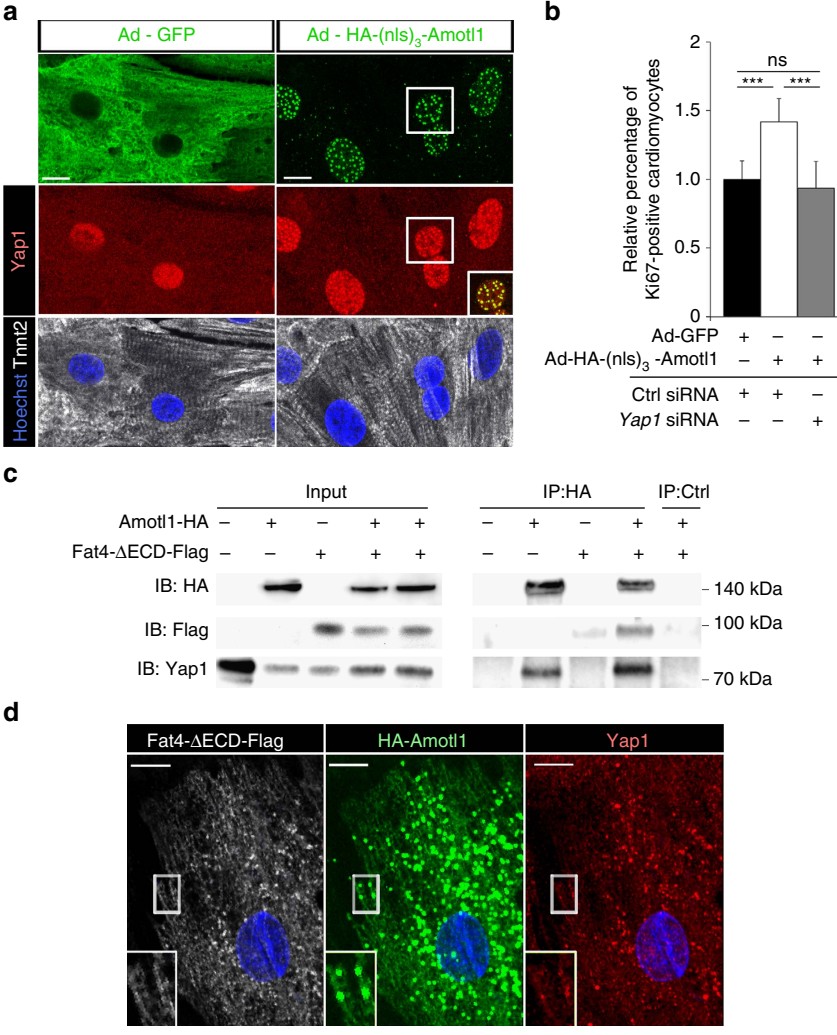

**Figure 6 | Amotl1 binds Fat4 and promotes cell proliferation.** (**a**) Immunodetection of Yap1 localization in primary cultures of cardiomyocytes infected with the indicated adenoviruses. The inset shows co-localization with Ad-HA-(nls)$_3$-Amotl1 in nuclear spots. (**b**) Increased percentage of proliferating Ki67-positive cardiomyocytes, as counted by flow cytometry from primary cell cultures infected with nuclear Amotl1 (Ad-HA-(nls)$_3$-Amotl1, $n = 8$), compared with controls (Ad-GFP, $n = 8$). Treatment with *Yap1* siRNA ($n = 8$) is inhibitory. ***$P < 0.001$ (analysis of variance). (**c**) Immunoprecipitation (IP) of Amotl1-HA from HEK293 cells transfected or not with Amotl1-HA or Fat4-ΔECD-Flag. IB, immunoblot with the indicated antibodies. (**d**) Co-localization of Amotl1-HA, Fat4-ΔECD-Flag and endogenous Yap1 in primary cultures of transfected cardiac cells. The inset shows an enlargement of the boxed region. Ctrl, control; ns, no significant difference. Scale bars: 10 μm.

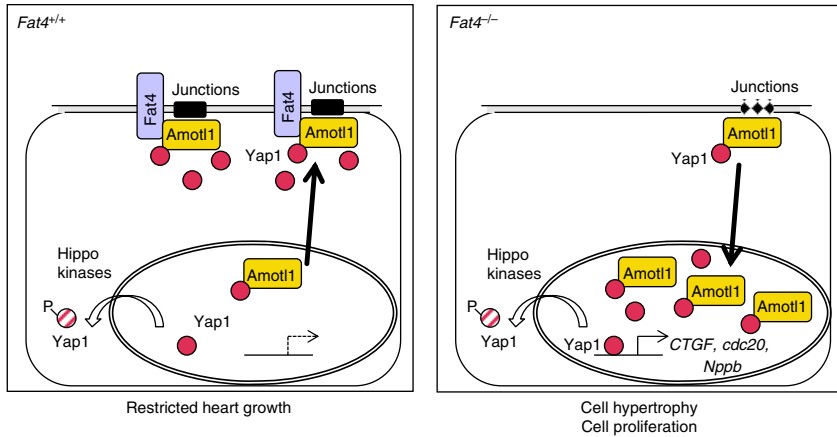

**Figure 7 | Model for the role of Fat4 in restricting heart growth.** Fat4 is required to organize cell junctions and sequester Amotl1, preventing excessive heart growth. In the absence of Fat4, Amotl1 is released and, in a complex with Yap1, translocates to the nucleus, bypassing the Hippo kinases. Resulting variations in gene expression promote proliferation and hypertrophy of cardiomyocytes, leading to excessive growth of the myocardium.

can occur independently of Yap1 phosphorylation[26]. In the heart, we observed that Angiomotin-like 1 (Amotl1), another member of the Angiomotin family, is predominantly expressed (Supplementary Fig. 4g). It is present near the membrane of cardiomyocytes, in clusters where Yap1 is also detected, which is compatible with a localization at cell junctions (Fig. 5c). Strikingly, Amotl1 was relocalized to the nucleus when Fat4 was absent in vivo (Fig. 5d,e) or downregulated in primary cultures of cardiomyocytes (Supplementary Fig. 3o). When we interfered with Amotl1 (Supplementary Fig. 2g–k) as well as Fat4 expression, we reversed the increased nuclear localization of Yap1 (Supplementary Fig. 3k,l) as well as the increased cell proliferation observed when Fat4 expression alone is down-regulated (Fig. 5f). Interference with Amotl1 expression alone shows that it is required for the proliferation of cardiomyocytes, with an effect dependent on the presence of Fat4. On the contrary, forcing Amotl1 to translocate to the nucleus, by addition of a nuclear localization signal, resulted in co-accumulation of Yap1 in the nucleus (Fig. 6a, Supplementary Fig. 3m–o) and stimulation of cardiomyocyte proliferation (Fig. 6b). Interference with Yap1 expression shows that the proliferative effect of nuclear Amotl1 is dependent on Yap1. Both Fat4 and Amotl1 are known to interact with the scaffold multi-PDZ domain protein, Mpdz, also known as Mupp1 (refs 27,28). We now show, in transfected HEK293 cells, that Amotl1 interacts physically with Fat4, as well as Yap1 (Fig. 6c), and that the three proteins co-localize in transfected cardiac cells (Fig. 6d). This is consistent with Amotl1 release from cell junctions when Fat4 is absent. These results identify Amotl1 as a modulator of the Hippo pathway in cardiomyocytes, and show that it is prevented from entering the nucleus by sequestration in a Fat4 complex, thus restricting Yap1 mediated tissue growth.

## Discussion

Our observations of Yap1 activity in Fat4 mutants, as well as the suppression of Fat4-dependent hyperproliferation by reduced Yap1 or Amotl1 expression, argue that Fat4 is an upstream regulator of Yap1 in the mouse heart, and that it triggers a non-canonical modulation of Hippo signalling. This pathway probably implicates a non-phosphorylated form of Yap1 bound to Amotl1 in the cytoplasm. When Amotl1 is not sequestered at cell junctions with Fat4, we show that it is an intermediate, which bypasses the Hippo kinases, to regulate the nuclear translocation of Yap1. Our results suggest that, depending on the available partners (for example, Fat4), Amotl1 has a role as an inhibitor or facilitator of Yap1. This is in agreement with previous reports showing contradictory roles of Angiomotin in different cell types[15,29,30]. Amotl1 may also directly contribute to the transcriptional activation of target genes, by analogy with Amot in the liver[15]. It remains to be seen whether Tead, a transcription factor that interacts with Yap1, is implicated in this context. The model that we propose is shown in Fig. 7. Amotl1 has no homologue in flies, which explains why the intracellular domain of Fat4 cannot rescue the growth phenotype of fat mutant flies[20]. The function of mouse Amotl1 is similar to that of Drosophila Expanded, a FERM-domain protein which requires Fat for its localization at the membrane[8] and which can directly sequester Yorkie out of the nucleus, independently of canonical Hippo signalling[31]. The mammalian homologue of Expanded, Frmd6, has lost the C-terminal domain of interaction with Hippo effectors, which supports an evolutionary switch in the regulation of Hippo signalling by Fat[19]. Although Fat signalling is implemented differently between mouse and fly, the function of this cadherin is well conserved, with a dual effect on tissue polarity[17] and also, as we now show, on tissue growth. The effect of Fat4 depends on the cellular context. In the heart, we show that Fat4 regulates tissue growth, rather than polarity. This has also been observed in the cortex[18], whereas in other organs, such as the kidney or the cochlea, Fat4 is a regulator of tissue polarity[16,17].

Our findings on Fat4 mutants uncover a mechanism that restricts heart growth at birth. Central to this mechanism is the adaptor protein Amotl1, which can shuttle from cell junctions to the nucleus, transporting the transcription co-factor Yap1. Whereas the Hippo pathway was shown to be required at embryonic stages of heart development[3,5], Fat4 is a later modulator exerting its role at birth. It remains to be established how the Fat4/Amotl1-dependent pathway is activated and what is its relative importance to regulate Yap1, in comparison with canonical Hippo signalling. Canonical Hippo signalling is also modulated by cell junctions in cardiomyocytes, where remodelling of the intercalated discs activates Hippo signalling, with pathological consequences leading to arrhythmogenic cardiomyopathy[32]. Our results showing the importance of Fat4 for the integrity of cell junctions is in agreement with the finding that Fat4 interacts with several junction proteins[12]. Fat4 mutant hearts display hypertrophy, in addition to increased cell proliferation. Although hypertrophy can potentially be induced by Yap1 (refs 2,26), other studies[4,5] would suggest that this is an indirect effect. Due to its positive effect on cardiomyocyte proliferation, Hippo signalling has been shown to be important for prolonging the regenerative potential of the mouse heart[6,7], which normally ceases during the first week after birth[33]. However, Yap1 is less efficient in promoting cardiomyocyte proliferation at postnatal stages than it is during development, which suggests that other factors block Yap1 activity at later stages. We now identify upstream regulators of Yap1 activity in the heart and anticipate that manipulating the Fat4 pathway will facilitate the reactivation of cardiomyocyte proliferation induced by phospho-resistant Yap1 (ref. 7) or Hippo kinase deficiency[6]. This has major therapeutic implications for the repair of the failing human heart.

## Methods

**Animal models.** The Fat4 mouse mutant line[17] was maintained in a 129S1 genetic background. Fat4 conditional mutants[17] were crossed to $Mesp1^{Cre/+}$ (ref. 34), $Wt1^{Cre/+}$ (ref. 35) lines or Yap1 conditional mutants[36] and backcrossed in the 129S1 genetic background. $Fat4^{-/-}$ mutants die at birth, whereas $Fat4^{flox/-}$; $Mesp1^{Cre/+}$ survive. Animal procedures were approved by the ethical committee of the Institut Pasteur and the French Ministry of Research. For histological analysis, hearts were excised, incubated in cold 250 mM KCl, fixed in 4% paraformaldehyde, embedded in paraffin in an automated vacuum tissue processor and sectioned on a microtome (10 μm). Male and female heart samples were mixed. For immunofluorescence studies, hearts were fixed in 0.5% paraformaldehyde, embedded in gelatine/sucrose, frozen in cold isopentane and sectioned on a cryostat (10 μm). For the quantification of tissue growth, images of sections stained with Haematoxylin Eosin or Hoechst were acquired. A polygonal mask was drawn in order to isolate the two ventricles from the atria. The green channel (with highest contrast) of the resulting image was inverted, thresholded and segmented using Connected Component analysis. The resulting regions were sorted, retaining the myocardial tissue and excluding blood speckles inside the ventricles, to compute the total area of the ventricles. The penetrance of the myocardial excessive growth was 75% ($n = 8$). Unless otherwise specified, the image analysis was done using the Icy software[37].

**RT–qPCR.** Complementary DNAs were reverse transcribed from RNAs extracted in TRIzol from cell cultures and isolated hearts using the Quanti-Tect kit (Qiagen) and Superscript II Reverse Transcriptase (Life Technologies) respectively. Quantitative PCR was carried out on a StepOne System (Life Technologies) using Fast Start SYBR Green Master (Roche). Quantification of gene expression was calculated as $R = 2^{\Delta Ct(\text{control-target})}$, with Gapdh used as a control. Primers were designed using the NCBI Primer-BLAST software. Primer sequences are listed in Supplementary Table 1.

**Primary cell culture.** Primary cultures were derived from newborn rat hearts. Cells were dissociated by treatment with collagenase in Tyrode buffer and, after centrifugation in a discontinuous Percoll gradient, cardiomyocytes were

preferentially recovered[38]. Primary cultures of cardiomyocytes, in DMEM—10% horse serum—5% fetal calf serum, were transfected with short interfering RNA (siRNA) at 10 nM using Lipofectamine RNAiMax with silencer-siRNA at 24 h and analysed at 72 or 96 h (Fig. 3c). Control (Ambion 4390843), Fat4 (siRNA-1: Ambion s172170, siRNA-2: CCUGUACCCUGAGUAUUGATT, siRNA-3 CCG UCCUUGUGUUUAACGUTT), Amotl1 (siRNA-1: AUCUCUACCAUUUGUUG GGTT, siRNA-2: GAGUAUCUCACGAGGCCUAUTT, siRNA-3: CAUCACAUGU CCCAGAAUATT), Yap1 (siRNA-1: Ambion s170200, siRNA-2: GUCAGAGAU ACUUCUUAAATT, siRNA-3: GGAGAAGUUUACUACAUAATT) siRNA were used. Efficiency of the interference was controlled by RT–qPCR. In the figures, Fat4 siRNA is Fat4-siRNA-1, Yap1 siRNA is a pool of siRNA-1 to 3 and Amotl1 siRNA is a pool of siRNA-1 to 3.

For flow cytometry analyses, cultures were dissociated to single cell suspensions by trypsin, fixed and permeabilized in eBioscience buffer. Proliferating cardiomyocytes were detected by immunostaining with primary antibodies against Tnnt2 (1/200, ab64623) and Ki67 (1/40, BD 556027) and counted on a BD LSRFortessa Cell Analyser cytometer. Gates were set according to isotype control antibodies (1/100, sc-3887; 1/40 BD556027). At least 900 cells were counted per condition. Alternatively, cardiomyocytes were exposed to EdU during 30 h and counted after immunofluorescence (at least 80 cells per condition). For overexpression experiments, cardiomyocytes were transfected using Lipofectamine 2,000 with Fat4-ΔECD-Flag (encoding Fat4 depleted for the extracellular domain and for the last C-terminal 297 nucleotides, derived from the constructs in ref. 12), HA-Amotl1 (ref. 28), Yap1-5SA (Addgene 27371) or control nuclear GFP (pCIG[39]) plasmids and analysed 24 h later. Alternatively, cardiomyocytes were infected with adenoviruses and analysed 24 h later, using control Ad-GFP[40] or newly generated HA-(nls)$_3$-Amotl1. It was cloned from human Amotl1 (ref. 41) in the Adeno-X Expression System 3 (Clontech).

**Immunofluorescence.** Immunofluorescence was performed with a standard protocol[21], using primary antibodies to acetylated tubulin (1/100, Sigma T6793), Actn2 (1/400, Sigma A7811), Amotl1 (1/50, Sigma HPA001196 and gift from D. Lallemand), Aurkb (1/300, BD 611082), non-phosphorylated (Ser[33/37]-Thr[41]) β-catenin (Ctnnb1, 1/100 Cell signalling 8814), Cav3 (1/100, BD 610420), Cdh2 (1/100, Ab12221), MF20 (1/100, DSHB), PH3 (1/300, ab32107), Pkp2 (1/50, Progen 651167), Scrib (1/100, sc-28737), Tnni3 (1/100, ab47003), Tnnt2 (1/200, ab64623), Vcl (1/100, Sigma F7053), Yap1 (1/100, sc-101199 and sc-15407), HA (1/200, Roche, 3F10), Flag (1/100, Sigma, F7425), GFP (1/500, Life Technologies A10262) and RFP (1/50, Chromotek 5f8), Alexa Fluor conjugated secondary antibodies (1/500) and Hoechst nuclear staining. Multi-channel 16-bit images were acquired with a Leica SP5 inverted confocal microscope and a 40 × /1.25 oil objective or with a Zeiss LSM 700 microscope and a 63 × /1.4 oil objective.

**Quantification of cell proliferation.** The PH3 channel was thresholded and segmented using Connected Component analysis, filtering objects under a minimum size of 16 μm[3] in order to eliminate non-specific signals. The myocardial volume of the multi-z scan was estimated by manually outlining the myocardial surface in the median Z-slice and computing the area. The total number of cardiomyocyte (α-actinin-positive) nuclei in the scan was estimated by manually counting the number of nuclei in a 200 pixels × 200 pixels window extending over all the Z-slices, and extrapolating to the total myocardial volume. More than 1,500 nuclei were counted per embryo.

**Quantification of cell size.** Images of cardiomyocyte transverse sections labelled with Caveolin3 (Cav3) were acquired systematically in the interventricular septum. Cell contours were drawn manually to compute cell area using ImageJ. At least 40 cells were counted per embryo.

**Quantification of protein localization.** The best in-focus Z-slice of the Hoechst channel was chosen for in vivo cells, whereas in vitro images were Z-projected. The analysis involved three image processing steps. In the first step, the myocardial (Tnnt2-positive) cells were segmented using Connected Component analysis applied after a Z-projection (sum) and thresholding of the Tnnt2 channel; alternatively, in vitro transfected cells were outlined manually. In the second step, the nuclei were segmented by thresholding after application of a Gaussian filter (in vivo), or by the 'Active Contours' plugin (in vitro). In the third step, the average intensity (total intensity divided by the area of the segmented object) of the protein of interest (PI) in the Tnnt2-positive cells (PItot) and in their nuclei (PInucl) was measured by multiplication of the PI channel with the respective binary images obtained in steps (1) and (2). Strong cells were defined as cells in which PInucl is higher than two standard deviations above the mean PInucl of control cells. The nuclear/cytoplasmic ratio was computed as: PInucl / PIcyto = PInucl/(PItot − PInucl).

**Quantification of the mosaicism of Cre expression.** The nuclei were first segmented by a K-Means classification of the histogram from the Hoechst channel (HK-Means plugin in ICY), extracting objects within a minimum/maximum size range. The average GFP and Tomato signals were then quantified in a box surrounding each nucleus (but excluding the nucleus itself). These two measures

were fed into a K-Means clustering algorithm implemented in Matlab (Pattern Recognition and Machine Learning Toolbox), with $k = 2$, effectively separating the GFP[high]/Tomato[low] from the GFP[low]/Tomato[high] cells.

**Immunoprecipitation and western blots.** HEK293 cells (Q-BIOgene AES0503) were transfected with Lipofectamine with the plasmids Amotl1-HA[28] and Flag-Fat4-ΔECD and cultured for 48 h. Proteins were extracted in a lysis buffer (150 mM NaCl, 5 mM EDTA, 10 mM Tris pH 7.5, 10% glycerol, 1% NP-40) in the presence of protease inhibitors. Immunoprecipitation of protein extracts was performed using a standard protocol based on magnetic beads coupled to bacterial protein G, an immunoglobulin-binding protein. One microgram of anti-HA antibody (Roche 3F10) per 500 μg of protein extracts was used. Proteins were eluted in Laemmli buffer. An isotype antibody (IgG sc38882) was used as a negative control of immunoprecipitation.

Proteins from cell cultures and isolated hearts were extracted for western blots in RIPA (150 mM NaCl, 5 mM EDTA, 50 mM Tris pH 7.4, 0.1% SDS, 1% NP-40) and NP40 (150 mM NaCl, 50 mM Tris pH 8, 1% NP-40) buffers, respectively, in the presence of protease and phosphatase inhibitors. Fractionation was performed in hypotonic (10 mM NaCl, 10 mM Tris, 1 mM DTT, 1 mM EDTA) and hypertonic (300 mM NaCl, 10 mM Tris, 1 mM DTT and 1 mM EDTA) buffers. The nuclear marker Phospho-histone H3 and the cytoplasmic marker Gapdh were used as controls of the fractionation. Equal amounts of proteins were separated on SDS–polyacrylamide gel electrophoresis and transferred to nitrocellulose or PDVF membranes. Proteins were detected with the primary antibodies Gapdh (1/20,000, Cell signalling 3683), HA (1/500, Roche 3F10), Thr[1079/1041] Phospho-Lats1/2 (1/1,000, Assay Biotech ref A8125), Lats1/2 (1/1,000, Bethyl A300-478A), Thr[183/180] Phospho-Mst1/2 (1/1,000, Cell signalling 3681), Mst1 (1/1,000, Cell signalling 3682), Mst2 (1/1,000, Cell signalling 3952), Ser[127] Phospho-Yap1 (1/1,000, Cell signalling 4911), Yap1 (1/1,000, Cell signalling 4912S) or Amotl1 (1/300, Sigma, HPA001196), PH3 (1/1,000, ab32107), followed by HRP-conjugated secondary antibodies (1/10,000, Jackson ImmunoResearch) and the ECL2 detection reagent. Protein quantification was obtained by densitometry analysis using a Typhoon laser scanner and normalized to Gapdh levels. Original un-cropped blots are shown in Supplementary Fig. 5.

**Image registration.** In order to obtain a full inferior view of the two ventricles at E10.5, confocal scans of the left ventricle, interventricular region and right ventricle were stitched together. The envelopes of the stitched images were computed by Active Mesh segmentation[42]. Ten such envelopes were used to compute an average envelope (referred to as the template), minimizing the deformation distances between the template and the envelopes, plus a residual mismatch cost. The metric distance was built on a group of smooth invertible deformations (that is, diffeomorphisms[43]). The axial data from each image were then transported through the deformation between the original envelope and the template, as described by the Jacobian matrix of the diffeomorphism (that is, the matrix of partial derivatives of the deformation, a three-dimensional generalization of the gradient). Using the polar part of the Jacobian was required to avoid improvement of the axial correlation.

**Quantification of tissue polarity.** Whole mount immunostaining was carried out on E10.5 isolated hearts using Scrib and Cadh2 antibodies to detect membranes and Aurkb antibody to detect cytoplasmic bridges. The nuclei and cytoplasmic bridges were segmented with the ICY software, using Filter Toolbox on the nuclear channel and Anisotropic Filter on the membrane channel, followed by subtraction of the two channels to clearly separate nuclei, and three-dimensional Active Mesh to extract the nucleus contour and centroid. On the Aurkb channel, Thresholder, Anisotropic Filter and Adaptive Histogram Equalization were applied. Sister cells were automatically detected using a scoring function of the distance from the bridge extremity and the direction of the bridge[21,44]. The axis of cell division was taken as the axis joining the two centroids of sister cell nuclei. For each genotype, at least three E10.5 embryonic hearts were registered, in order to pool the axial data on a common template. The planar component of each axis of cell division was extracted by projection over the template envelope. The threshold eigenvalue for each region size, $E_{1(5\%)}$, which was obtained by a bootstrap method[21], was calculated before and after the diffeomorphic transport of the axes, and the highest value was retained to compensate for any spurious improvement of the alignment due to the transport. Contour maps of axial coordination were produced by selecting the region, containing at least 50 axes, with the highest eigenvalue $E_1$ (core region), listing all regions that both included the core region and had an eigenvalue $E_1 > E_{1(5\%)}$, and drawing these regions on the template, with contour values equal to the ratio $E_1/E_{1(5\%)}$.

**Electron microscopy.** Neonate hearts were dissected in cold Krebs buffer without calcium, and fixed open with 2% glutaraldehyde in cacodylate buffer (Na Cacodylate 150 mmol l$^{-1}$, CaCl2 2 mmol l$^{-1}$, pH 7.3). The left ventricular papillary muscles were excised and fixed again in 2% gluteraldehyde in cacodylate buffer, post-fixed in 1% OsO4, contrasted in 1% uranyl acetate, dehydrated and embedded into Durcupan. Ultrathin (58–60 nm) longitudinal sections were cut by Power-Tome MT-XL (RMC/ Sorvall, USA) ultramicrotome, placed on copper slot

grids covered with formwar and stained with lead citrate. The sections were examined in a JEM 2000FX (Jeol, Japan) electron microscope and recorded using a Gatan DualVision 300W CCD (charge-coupled device) camera (Gatan Inc., USA).

**ImageStream.** P14 hearts were collected, minced and flash frozen in liquid nitrogen[45]. The defrosted tissue was fixed in 4% paraformaldehyde, digested with $3 \text{ mg ml}^{-1}$ collagenase type II in HBSS and filtered using a 100 µm cell-strainer. Staining of isolated cells was performed with the BD Cytofix/Cytoperm Fixation/Permeabilization Kit, using anti-sarcomeric α-actinin (1/600, Sigma) and DRAQ5 nuclear stain. Data acquisition was performed using an ImageStreamX cytometer with INSPIRE software (Amnis). Files were collected with a cell classifier applied to the brightfield channel to capture events larger than 100 µm. At least 23,000 cell events were acquired for each sample and all images were captured with the $40 \times$ objective. Data analysis was performed with IDEAS software (v6.0, Amnis). Images were compensated using a matrix generated by single-stained samples acquired with identical laser settings in the absence of brightfield illumination. The analysis was restricted to in-focus single cells and to intact cardiomyocytes, selected as actinin and DRAQ5 double positive. An object mask was created on the brightfield channel and the aspect ratio was defined as the ratio between the minor and major cell axis. The number of nuclei per cell was assessed using the DRAQ5 images, in at least 350 cells per heart.

**Statistics.** Sample size was chosen in order to ensure a power of at least 0.8, with a type I error threshold of 0.05, in view of the minimum effect size that was looked for. The sample size was calculated using the observed variance of the wild-type mice for the phenotype considered. Sample outliers were excluded according to the Thompson Tau test. The experiments were not randomized and the investigators were not blinded to allocation during experiments and outcome assessment.

Comparisons of centre-values were done on either the average or the geometrical mean when ratios were compared. An analysis of variance was systematically calculated when more than two centre-values were compared, and Tukey-Kramer's two-sided test was used for the assessment of bilateral significance. Otherwise, a Student two-sided test was used. When $n > 10$, normality was checked by a Kolmogorov–Smirnov test or by visualization of the distribution. When test was not positive, a Wilcoxon U-test was used.

For quantitative data, the number of observations and replications of the experiments are provided in Supplementary Table 2. When $n < 5$, the figures display all data points. For data shown as representative images, the number of replications of the experiments are: 1 experiment (Fig. 5b [$n = 1$ $Fat4^{+/+}$, 2 $Fat4^{+/-}$ and 2 $Fat4^{-/-}$ hearts], 6a [3 cultures], Supplementary Fig. 3o [2 cultures], Supplementary Fig. 4e [$n = 2$ $Fat4^{+/+}$ and 3 $Fat4^{-/-}$ hearts]), 2 experiments (Fig. 3b [$n = 3$ $Fat4^{+/+}$ and 3 $Fat4^{-/-}$ hearts], 5a [$n = 2$ $Fat4^{+/+}$ and 2 $Fat4^{-/-}$ hearts], 5c [$n = 3$ $Fat4^{+/+}$ hearts], Supplementary Fig. 2b [$n = 4$ $Fat4^{+/+}$, 2 $Fat4^{+/-}$ and 2 $Fat4^{-/-}$ hearts], Supplementary Fig. 2n [$n = 3$ hearts]), 3 experiments (Figs 4e and 6c,d), 4 experiments (Fig. Supplementary Fig. 3a), 50 litters (Fig. 1a), 3 litters (Supplementary Fig. 2d), 1 litter at E14.5, 3 litters at E16.5, 2 litters at E18.5 (Supplementary Fig. 4b).

**Data availability.** The authors declare that all data supporting the findings of this study are available within the article and its Supplementary Information files or from the corresponding author upon reasonable request.

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

## Acknowledgements

We are very grateful to M. Buckingham for support, encouragement and advice. C. Cimper, J. Leloup, T. Lucchesi, R. Coelho and L. Novota provided technical assistance. We thank G. Odelin, L. Fiette and F. Bourgade for advice, P.H. Commere for assistance with flow cytometry, P. Švec and R. Ventura for assistance with electron microscopy, D. Lallemand for discussions and sharing antibodies, M. Adachi and A. Schmitt for the gift of plasmids. Images were acquired at the imaging platforms of the Institut Pasteur and the SFR Necker. This work was supported by grants from VEGA [2/0110/15] to M.N. and from the Institut Pasteur [PTR335], ANR [11-JSV2-00601], FRM [DPC20111122997], CNRS [PEPS BMI] to S.M.M. C.V.R. was funded by the MESR and ARC, N.D. and L.K. by the Fondation Lefoulon-Delalande, L.K. by the Institut Servier and S.M.M. is an INSERM research scientist. T.P.R. is funded by the FCT [SFRH/BPD/80588/2011] and INEB [NORTE-07-0124-FEDER-000005] to P. Pinto-do-Ó.

## Author contributions

C.V.R. performed the *in vivo* work, N.D. performed the work on cell cultures, J.-F.L.G. performed the polarity analysis and supervised the quantitative work, M.N. performed the electron microscopy, T.P.R. performed the ImageStream analysis, L.G. and L.K. performed the cell fractionation, S.P., A.D. and J.-C.O.-M. designed the algorithms of image segmentation, N.C. and A.T. designed the algorithms of image registration, C.B. and H.M.N. provided the *Fat4* mutant lines, reagents and expertise, S.M.M. contributed to the experimental work, conceived and directed the project and wrote the manuscript with input from all authors.

## Additional information

**Competing financial interests:** The authors declare no competing financial interests.

