## [Peer review file · Nature Communications]

Editorial Note: This manuscript has been previously reviewed at another journal that is not operating a transparent peer review scheme. This document only contains reviewer comments and rebuttal letters for versions considered at Nature Communications. Mentions of prior referee reports have been redacted.

Reviewers' comments:

Reviewer #1 (Remarks to the Author):

I think this is an important paper that makes a number of very strong contributions. Certainly, every interaction is not mapped at a biochemical level, but no paper before this one can make that claim.

Reviewer 3 is hostile and many of his/her comments are rhetorical and in a distinctly negative fashion (".. so their refusal to carry out these crosses is difficult to understand..") - in particular a single potential weakness is seized upon or called a "fatal weakness" without reference to the balance of evidence. This is inappropriate and I believe the editors should take a strong position on this and guide the authors from here as to what comments are reasonable to address experimentally or in the text. Reviewer 3 seems disproportionately dismissive of the in vitro data in cardiomyocytes, which are carefully done with thorough quantifications and provide strong support to the model. With respect to the cited small changes and/or larger error bars in level of Ki67, pHH3 and proportion of cardiomyocytes, it should be pointed out that small changes in the percentage of cells transiting S or G2/M phases at a specific time point, translates over several days or a fraction of lifetime to the addition of tens of thousands or hundreds of thousands of cardiomyocytes. For example, the increase in Fat4 mutants in proportion of CMs in Fig. 2c seems small but is around 4%: the cardiac ventricles at birth contain about 1 million cardiomyocytes and this increases to about 1.5 million in adults. A 4% increase in the number of CMs at birth (approximately the period that Fat4 is active on the Hippo pathway) due to Fat4 mutation represent about 40,000 new cardiomyocytes. This may be an underestimate since the measurement was "proportion" and new vessels and stroma will be added as new CMs are added. Moreover, the large heart phenotype is due in part to cardiomyocyte division and in large part also to induction of hypertrophy. Some clarification of these points in the text might would be for readers.

With respect to the Mesp1-Cre and the Wt1-Cre crosses, the authors have done what they can with the mice available. These crosses demonstrate that cardiac restricted expression conferred by the Mesp1 regulatory elements recapitulate the increase in relative ventricular area (noting that this is a combination of proliferation and hypertrophy) seen in germline Fat4 mutants, albeit that the effect is less striking. Formally, this could indicate some contribution from a systemic or non-cardia influence, but it is more likely to be due to the limited and/or variable efficacy of Cre alleles on the Fat4 locus. This sort of thing is common. While the rescue with Yap Flox/+ does not reach significance, calling it a trend is I think allowable, and importantly there is a significant rescue of the number of pHH3+ and the percent CMs in this setting. The delta for the latter is 5-6% (see my comments on how this translates to number of CMs above). If these data are significant, reviewer 3 should identify what possible flaws could invalidate the conclusion. Wt1Cre will mark the descendants of the epicardium and possibly additionally endothelial cells: that is a large proportion of the stromal and vascular/perivascular elements (non-CM) of the heart. The data from this cross does NOT replicate the findings on the germline or Mesp1-Cre Fat4 deletions. This is all valuable data with a cell autonomous role for Fat4 in CMs strongly supported by the work in cultured CMs.

With respect to point 14 in Reviewer 3's latest comments, he/she makes an interesting point. In the absence of any other data or understanding of the nature of the players in this pathway, it would only be possible to take the neutral view that one pathway modulates the effects of the other. It does not establish epistasis. But this is not a defect in the data which is incredibly valuable. In fact, these data support the authors' concept in two ways. They show that the impact of Fat4 knockdown on relative percent of Ki67+ cells is ameliorated by knockdown of Amotl1, as expected if Amotl1 is downstream of Fat4 in the way envisaged, AND that the strong effects of Amotl1 knockdown alone are dependent on the presence of Fat4. Moreover, looking at the broader context, Fat4 is a membrane adhesion protein while Amotl1 is a multifunctional protein with roles at the membrane and in the nucleus and for which there is evidence (albeit in vitro) for an interaction with both Fat4 and Yap1. Thus, I think it is reasonable for the authors to state their conclusion in the way they do. The comment in point 9 about whether the strong effects of nuclear Amotl1 on Ki67+ cells are serial or parallel are I suppose valid but rather off the point: all networks are complex with multiple cross-regulatory interactions and feedbacks and parallelism or serialism are not established in any epistasis assay. This data adds to the model in a compelling way. If Amotl1 becomes more compellingly localised to the nucleus in Fat4 mutants, in which CM proliferation is stimulated, then the fact that a nuclear localising form of Amotl1 stimulated proliferation supports the "serial" model in the absence of evidence to suggest a parallel model.

Please let me know if this addresses the points you were interested in. I strongly support the paper for Nature Communications.

Reviewer #2 (Remarks to the Author):

The authors report that Fat4 regulates heart growth by sequestering YAP1 out of the nucleus via Amotl1. Overall this conclusion is novel and important and the authors have modified the manuscript in response to prior critiques. However the data supporting the proposed mechanism continues to be somewhat thin and could be strengthened.

1. There is a discrepancy of the measurement of relative ventricular area between control and mutant in Fig. 1 (Fat4 +/-, rel vent area ~ 1.7) compared to Fig. 3 (Fat4 fl/- Mesp1 Cre/+, rel vent area ~ 1.2). Does this mean that most of the effect of Fat4 on vent size is not in the Mesp1Cre domain? Absolute measure of HW and HW/BW is the typical way to report on change in heart size and would allow more meaningful comparison between genetic models.
2. Many of the key measurements are based on measurements from immunofluorescent images. While differences between groups are significant, they are also small. Independent validation by other methods, such as biochemical fractionation of nuclear and cytoplasmic fractions, would be welcome.
3. Figure 4C putatively shows Yap1 and Amotl1 both localizing near the cell junction of cardiomyocytes. This localization remains poorly demonstrated. Do the authors mean to imply that these proteins are co-localizing? This should be shown by co-immunostaining. Co-staining with cell junction markers would also be helpful. In Figure 4b, the finding of reduced gap junctions could be better supported by performing Cx43 IF staining.
4. In Figure 4E, Amotl1 N/C signal of het and ko are similar, whereas the het and ko phenotypes are not. Please explain.
5. In Fig. 4F-H, please relate the change in CM proliferation in each group to the change in localization of Fat4, Amotl1, and YAP. For example, by subcellular fractionation and western blotting.

Reviewer #3 (Remarks to the Author):

In this revised MS, the authors failed to offer convincing evidence to support their thesis. The key concerns of Rev2 and Rev3 remain unaddressed.

The paper shows the phenotype of fat4 knockout, but it does not go much further. It is not easy for me to follow and derive the conclusions stated in title, abstract and final diagram. Even if we stretch things, the title is massively misleading (implying an opposite message to their own interpretation of the data, that is that AMOTL1 is a nuclear factor playing positive role in something that may overlap with what YAP does in cardiomyocytes).

- 1) fig1 there is a quite impressively overgrown heart, in $-/-$, but not in $+/-$. Same in 2b, and 3a (although there is a huge standard deviation casting doubt on YAP target gene upregulation).
 - 2) fig.2 gene expression: the phenotype is in $-/-$ = to $+/-$. What is the significance of those RNA changes (given the distinct phenotype-genotype connections shown in fig1 and 2b)?
 - 3) fig3G. one single fat4 sirna is upregulating Ki67 from 1 to 1.3. The effect is minor. There is no criteria/data on how Ki67 positivity was scored.
 - 4) Fig 3h and i. by using the mesp-Cre driver they cannot recapitulate what shown in the total body embryonic KO of Fig.1. this was part of Rev2 concern. What is going on? The effect, if anything this is truly minor: There is a 20% increase, perhaps. Again, the statistical value of this change is questionable, as the criteria of these calculations remains unclear (but see point 6 below).
 - 5) In the same vein, when I compare Mesp-Cre to Wt1Cre (whole cardiac vs cardiac-but-not-cardiomyocytes) in S2 and S4, frankly, I see no difference. Moreover, they claim in their response to rev.2 that they have been unable to provide positive and direct data on the cardiomyocyte knockout, that this would have been important to solidify their claims and to support their in vitro data.
- In sum, they cannot exclude that the Fat4 phenotype shown in fig1 may be the product of more systemic effects or of a requirement in cells that are not cardiomyocytes. Tissue-specific chalone? Hormones?
- In addition, the phenotypic description of the Mesp1-Cre and Wt1-Cre mice is very poor.
- 6) In Fig. 2i, there is no rescue between the red and orange bar (YAP flox). So either "mutants" (?) is not a real phenotype (and then, again, we are left with no idea of what is going on in Figure 1) or this is a real, although mild phenotype, but this is YAP independent. In either scenario, the paper displays a fatal weakness.
 - 7) AMOT, at the center of this story, appears only at the end, in Figure 4d-f. they show that AMOT is essential for Ki67. This can be quite indirect.
 - 8) Why and How is this showing any YAP connection? Any consistent modification of YAP or TAZ nuclear localization? Or of YAP targets? this is not shown.
 - 9) Fig4h. Overexpressed Amot (with 3 nuclear tags) has a 30% increased Ki67, in turn "rescued" by YAP inactivation. Why are this assumed to be serial, rather than parallel regulations? There is no way to distinguish this from what shown here.
 - 10) As previously requested, the interaction with overexpressed protein is suggestive at best, but not acceptable by itself. Endogenous proteins have not been provided during revision. How do we know that those associations are real?
 - 11) What is the rationale of jumping from an siRNA data (and see point 13) to overexpressing a nuclear AMOT1?

12) What are the consequences of nuclear AMOT for YAP, since YAP does not change its nuclear localization after nuclear-Amot overexpression (Fig 4d),

13) Fig 4f: how can lane 2 and 3 be statistically significant?

14) Fig 4f: the interpretation is really biased. I may say, by comparing lane 3 and 4, that it is actually fat4 siRNA that rescues Amot siRNA. So, I may conclude that Fat4 is downstream of Amot11, and not viceversa. There is no redundant evidence (let alone YAP specificity) supporting the view that what we are looking at is the model they are portraying.

In sum, even if we do sidestep the request of Rev.3 to show AMOTL1 genetic requirement (that I would consider mandatory at this stage in light of the complexity of the various AMOT functions so far reported), the paper is anyway not showing any rescue with YAP conditional knockouts (there is actually a negative data, see point 6 above). We are left with a fat 4 whole embryonic phenotype, and siRNA data showing in vitro results that are quantitatively minor and open to different interpretations.

There is no validated connection between the Fat4 phenotype shown in Fig1 and YAP; or between FAT and AMOTL1 (and these mice are available so their refusal to carry out these crosses is difficult to understand) or between AMOT11 (of unknown relevance in the heart) and YAP.

The problem of the MS is not on mechanisms but on the soundness of the data and gene-interactions provided (still at the functional level).

The final drawing goes well beyond the data.

In their response to the reviewers:

"In the heart we show that Amot11 has an inhibitory role in the presence of Fat4, by sequestering Yap1 out of the nucleus"

this is shown nowhere in the paper. There is an siRNA for AMOT11 opposing basal ki67 in cultured cardiomyocytes.

Response to reviewers

Reviewer #1

I think this is an important paper that makes a number of very strong contributions. Certainly, every interaction is not mapped at a biochemical level, but no paper before this one can make that claim.

Reviewer 3 is hostile and many of his/her comments are rhetorical and in a distinctly negative fashion (".. so their refusal to carry out these crosses is difficult to understand..") - in particular a single potential weakness is seized upon or called a "fatal weakness" without reference to the balance of evidence. This is inappropriate and I believe the editors should take a strong position on this and guide the authors from here as to what comments are reasonable to address experimentally or in the text. Reviewer 3 seems disproportionately dismissive of the *in vitro* data in cardiomyocytes, which are carefully done with thorough quantifications and provide strong support to the model. With respect to the cited small changes and/or larger error bars in level of Ki67, pHH3 and proportion of cardiomyocytes, it should be pointed out that small changes in the percentage of cells transiting S or G2/M phases at a specific time point, translates over several days or a fraction of lifetime to the addition of tens of thousands or hundreds of thousands of cardiomyocytes. For example, the increase in Fat4 mutants in proportion of CMs in Fig. 2c seems small but is around 4%: the cardiac ventricles at birth contain about 1 million cardiomyocytes and this increases to about 1.5 million in adults. A 4% increase in the number of CMs at birth (approximately the period that Fat4 is active on the Hippo pathway) due to Fat4 mutation represent about 40,000 new cardiomyocytes. This may be an underestimate since the measurement was "proportion" and new vessels and stroma will be added as new CMs are added. Moreover, the large heart phenotype is due in part to cardiomyocyte division and in large part also to induction of hypertrophy. Some clarification of these points in the text might would be for readers.

We thank the reviewer for his strong support. We have now clarified the text accordingly. The text acknowledges in 3 instances that Fat4 affects both cell proliferation and cell hypertrophy (summary, results, discussion).

With respect to the *Mesp1-Cre* and the *Wt1-Cre* crosses, the authors have done what they can with the mice available. These crosses demonstrate that cardiac restricted expression conferred by the *Mesp1* regulatory elements recapitulate the increase in relative ventricular area (noting that this is a combination of proliferation and hypertrophy) seen in germline Fat4 mutants, albeit that the effect is less striking. Formally, this could indicate some contribution from a systemic or non-cardia influence, but it is more likely to be due to the limited and/or variable efficacy of Cre alleles on the Fat4 locus. This sort of thing is common. While the rescue with *Yap Flox/+* does not reach significance, calling it a trend is I think allowable, and importantly there is a significant rescue of the number of pHH3+ and the percent CMs in this setting. The delta for the latter is 5-6% (see my comments on how this translates to number of CMs above). If these data are significant, reviewer 3 should identify what possible flaws could invalidate the conclusion. *Wt1Cre* will mark the descendants of the epicardium and possibly additionally endothelial cells: that is a large proportion of the stromal and vascular/perivascular elements (non-CM) of the heart. The data from this cross does NOT replicate the findings on the germline or *Mesp1-Cre* Fat4 deletions. This is all valuable data with a cell autonomous role for Fat4 in CMs strongly supported by the work in cultured CMs.

We have now quantified the number of cells targeted by the *Mesp1-Cre* allele as 70% of cardiac cells (Supplementary figure 2n) and highlighted in the text that this potentially explains why the phenotype of the conditional *Fat4* mutants is milder than that of the constitutive mutants.

With respect to point 14 in Reviewer 3's latest comments, he/she makes an interesting point. In the absence of any other data or understanding of the nature of the players in this pathway, it would only be possible to take the neutral view that one pathway modulates the effects of the other. It does not establish epistasis. But this is not a defect in the data which is incredibly valuable. In fact, these data support the authors concept in two ways. They show that the impact of Fat4 knockdown on relative percent of Ki67+ cells is ameliorated by knockdown of *Amotl1*, as expected if *Amotl1* is downstream of Fat4 in the way envisaged, AND that the strong effects of *Amotl1* knockdown alone are dependent on the presence of Fat4. Moreover, looking at the broader context, Fat4 is a membrane adhesion protein while *Amotl1* is a multifunctional protein with roles at the membrane and in the nucleus and for which there is evidence (albeit *in vitro*) for an interaction with both Fat4 and *Yap1*. Thus, I think it is reasonable for the authors to state their conclusion in the way they do. The comment in point 9 about whether the strong effects of nuclear *Amotl1* on Ki67+ cells are serial or parallel are I suppose valid but

rather off the point: all networks are complex with multiple cross-regulatory interactions and feedbacks and parallelism or serialism are not established in any epistasis assay. This data adds to the model in a compelling way. If Amotl1 becomes more compellingly localised to the nucleus in Fat4 mutants, in which CM proliferation is stimulated, then the fact that a nuclear localising form of Amotl1 stimulated proliferation supports the "serial" model in the absence of evidence to suggest a parallel model.

We fully agree with the reviewer. We have now clarified the text accordingly and provide novel biochemical evidence of (1) the increased nuclear localisation of Yap1 upon Fat4 knock-down or accompanying Amotl1 forced nuclear translocation (Supplementary figure 3k-n), (2) the increased nuclear localisation of Amotl1 upon Fat4 knock-down (Supplementary figure 3o).

Please let me know if this addresses the points you were interested in. I strongly support the paper for Nature Communications.

--

Reviewer #2

The authors report that Fat4 regulates heart growth by sequestering YAP1 out of the nucleus via Amotl1. Overall this conclusion is novel and important and the authors have modified the manuscript in response to prior critiques. However the data supporting the proposed mechanism continues to be somewhat thin and could be strengthened.

1. There is a discrepancy of the measurement of relative ventricular area between control and mutant in Fig. 1 (Fat4 +/-, rel vent area ~ 1.7) compared to Fig. 3 (Fat4 fl/- Mesp1 Cre/+, rel vent area ~ 1.2). Does this mean that most of the effect of Fat4 on vent size is not in the Mesp1Cre domain? Absolute measure of HW and HW/BW is the typical way to report on change in heart size and would allow more meaningful comparison between genetic models.

We have now quantified the number of cells targeted by the Mesp1-Cre allele as 70% of cardiac cells (Supplementary figure 2n) and highlighted in the text that this potentially explains why the phenotype of the conditional *Fat4* mutants is milder than that of the constitutive mutants.

As we explained in our first round of revision, we have quantified the heart weight to tibia length ratio at P0 in *Fat4*^{-/-} mutants. However, we did not find any significant difference. Given the small size of neonate hearts, it is difficult to measure weights precisely. We found that measuring the area of the ventricles in histological sections is more reliable, i.e. provides a measure with less variance for control samples (the normalised standard deviation was 11.5%, n=7, for histological measures instead of 18%, n=5 for heart weight ratios).

2. Many of the key measurements are based on measurements from immunofluorescent images. While differences between groups are significant, they are also small. Independent validation by other methods, such as biochemical fractionation of nuclear and cytoplasmic fractions, would be welcome.

5. In Fig. 4F-H, please relate the change in CM proliferation in each group to the change in localization of Fat4, Amotl1, and YAP. For example, by subcellular fractionation and western blotting.

We have now performed fractionated Western Blots to further evaluate the nuclear/cytoplasmic levels of Amotl1 and Yap1 in primary cultures of cardiomyocytes presented in Fig. 4F-H. Of course, protein levels cannot be measured in knock-down conditions (see supplementary figure 2h). As we explained before, we do not have the appropriate antibody to detect endogenous Fat4. Thus, we provide novel biochemical evidence of (1) the increased nuclear localisation of Yap1 upon Fat4 knock-down or accompanying Amotl1 forced nuclear translocation (Supplementary figure 3k-n), (2) the increased nuclear localisation of Amotl1 upon Fat4 knock-down (Supplementary figure 3o).

3. Figure 4C putatively shows Yap1 and Amotl1 both localizing near the cell junction of cardiomyocytes. This localization remains poorly demonstrated. Do the authors mean to imply that these proteins are co-localizing? This should be shown by co-immunostaining. Co-staining with cell junction markers would also be helpful. In Figure 4b, the finding of reduced gap junctions could be better supported by performing Cx43 IF staining.

We showed co-localisation in cell cultures transfected with tagged constructs (Fig. 4k), as well as in co-immunoprecipitation. In vivo, we are limited by the Yap1 and Amot1 antibodies, which are both raised in the Rabbit and thus not amenable to co-localisation studies. We show in vivo, that they have a similar subcellular localisation, with co-labelling of the junction marker vinculin. Yap1 and Amot1 are known to be strong partners.

We have now performed immunostaining with Cx43. This did not reveal any qualitative difference in the distribution of Cx43 spots between control and mutant hearts. It is unclear why the subcellular localisation of Cx43 is unperturbed, when we detected defects in the structure of gap junctions by electron microscopy. We have thus decided to remove our claim on the absence of gap junctions, to keep only our strong conclusion on abnormal intercalated discs, consistently observed both by immunostaining and electron microscopy.

4. In Figure 4E, Amot1 N/C signal of het and ko are similar, whereas the het and ko phenotypes are not. Please explain.

After the first round of revision, we added a sentence to make this point in the results “Heterozygotes also show transcript upregulation, although they do not have a detectable heart phenotype, indicating compensation at the level of the proliferation gene network dependent on Fat4 dosage”. It is difficult to expect a linear relationship between the expression of a few genes at a single stage and a phenotype which integrates 18 days of development and networks of gene regulation. Compensation mechanisms may intervene in heterozygotes, such that impaired gene expression does not lead to the phenotype. This is in keeping with the robustness of developmental pathways “the genotype can, as it were, absorb a certain amount of its own variation without exhibiting any alteration in development” (Waddington, 1942). There are multiple potential sources of robustness, including modularity and redundancy, network architecture, and feedback control (Stelling et al., 2004). The same rationale would also apply to protein signals.

--

Reviewer #3

In this revised MS, the authors failed to offer convincing evidence to support their thesis. The key concerns of Rev2 and Rev3 remain unaddressed.

The paper shows the phenotype of fat4 knockout, but it does not go much further. It is not easy for me to follow and derive the conclusions stated in title, abstract and final diagram. Even if we stretch things, the title is massively misleading (implying an opposite message to their own interpretation of the data, that is that AMOTL1 is a nuclear factor playing positive role in something that may overlap with what YAP does in cardiomyocytes).

Amot1 becomes nuclear when Fat4 is absent. Therefore, the title provides the conclusion in the wild-type (not mutant) situation.

1) fig1 there is a quite impressively overgrown heart, in $-/-$, but not in $+/-$. Same in 2b, and 3a (although there is a huge standard deviation casting doubt on YAP target gene upregulation).

In vivo measurements are more difficult compared to in vitro experiments : fewer samples are available and higher standard deviations are observed. This is why we have performed careful statistical analyses and combined different types of observations to draw conclusions.

2) fig.2 gene expression: the phenotype is in $-/-$ = to $+/-$. What is the significance of those RNA changes (given the distinct phenotype-genotype connections shown in fig1 and 2b)?

After the first round of revision, we added a sentence to make this point in the results “Heterozygotes also show transcript upregulation, although they do not have a detectable heart phenotype, indicating compensation at the level of the proliferation gene network dependent on Fat4 dosage”. It is difficult to expect a linear relationship between the expression of a few genes at a single stage and a phenotype which integrates 18 days of development and networks of gene regulation. Compensation mechanisms may intervene in heterozygotes, such that impaired gene expression does not lead to the phenotype. This is in keeping with the robustness of developmental pathways “the genotype can, as it were, absorb a certain amount of its own variation without exhibiting any alteration in development” (Waddington, 1942). There are multiple potential sources of robustness, including modularity and redundancy, network architecture, and feedback control (Stelling et al., 2004).

3) fig3G. one single fat4 sirna is upregulating Ki67 from 1 to 1.3. The effect is minor. There is no criteria/data on how Ki67 positivity was scored.

We explain in the Methods section (Primary Cell Culture) « For flow cytometry analyses, cultures were dissociated to single cell suspensions by trypsin, fixed and permeabilized in eBioscience buffer. Proliferating cardiomyocytes were detected by immunostaining with primary antibodies against Tnnt2 (ab64623) and Ki67 (BD 556027) and counted on a BD LSRFortessa Cell Analyzer cytometer. Gates were set according to isotype control antibodies (sc-3887). At least 900 cells were counted per condition. Alternatively, cardiomyocytes were exposed to EdU during 30h and counted after immunofluorescence (at least 80 cells per condition) ». This is illustrated by supplementary figure 2j and l-m. In supplementary figure 2k, we show that treatment with two other siRNA against Fat4 also results in an increased number of Ki67 positive cells.

4) Fig 3h and i. by using the *mesp*-Cre driver they cannot recapitulate what shown in the total body embryonic KO of Fig.1. this was part of Rev2 concern. What is going on? The effect, if anything this is truly minor: There is a 20% increase, perhaps. Again, the statistical value of this change is questionable, as the criteria of these calculations remains unclear (but see point 6 below).

5) In the same vein, when I compare *Mesp*-Cre to *Wt1*Cre (whole cardiac vs cardiac-but-not-cardiomyocytes) in S2 and S4, frankly, I see no difference. Moreover, they claim in their response to rev.2 that they have been unable to provide positive and direct data on the cardiomyocyte knockout, that this would have been important to solidify their claims and to support their in vitro data.

In sum, they cannot exclude that the *Fat4* phenotype shown in fig1 may be the product of more systemic effects or of a requirement in cells that are not cardiomyocytes. Tissue-specific chalone? Hormones?

In addition, the phenotypic description of the *Mesp1*-Cre and *Wt1*-Cre mice is very poor.

As pointed by Reviewer 1, small changes in the percentage of cells transiting S or G2/M phases at a specific time point, translates over several days or a fraction of lifetime to the addition of tens of thousands of cardiomyocytes. An increase by 5% in the number of cardiomyocytes at birth due to *Fat4* mutation represents about 50,000 new cardiomyocytes.

The *Mesp1*-Cre and *Wt1*-Cre alleles have been described in previous publications (Saga et al., 1999 and 2000; Wessels et al., 2012). We have now quantified the number of cells targeted by the *Mesp1*-Cre allele as 70% of cardiac cells (Supplementary figure 2n) and clarified in the text that this potentially explains why the phenotype of the conditional *Fat4* mutants is milder than that of the constitutive mutants.

The reviewer seems to ignore the timescale for generating mouse crosses in a given genetic background (129S1), as it would be required to generate a novel *Fat4* conditional mutant with a novel Cre driver (2 years of crosses). During the first round of revision, we have added data about conditional mutants. We have carefully quantified, and improved during the second round of revision, the histological and immunostaining data in the conditional mutants, as shown in Fig3 h-k and supplementary figure 2d-f. The phenotypes in *Mesp*-Cre and *Wt1*Cre conditional mutants are not the same. We have completed these observations by in vitro data supporting a cell autonomous role for *Fat4* in cardiomyocytes.

We have not tested the hypothesis that hormones may influence the phenotype. This is beyond the scope of the manuscript.

6) In Fig. 2i, there is no rescue between the red and orange bar (YAP flox). So either "mutants" (?) is not a real phenotype (and then, again, we are left with no idea of what is going on in Figure 1) or this is a real, although mild phenotype, but this is YAP independent. In either scenario, the paper displays a fatal weakness.

Although there is no significant rescue, the p value is very close to significance, which points to a strong tendency towards a rescue.

7) AMOT, at the center of this story, appears only at the end, in Figure 4d-f. they show that AMOT is essential for Ki67. This can be quite indirect.

8) Why and How is this showing any YAP connection? Any consistent modification of YAP or TAZ nuclear localization? Or of YAP targets? this is not shown.

We have extensively studied Yap1 nuclear localisation (Figure3 b-f, n-o) and targets (Fig. 2d, 3a). We now provide novel biochemical evidence, by fractionated Western blot, of (1) the increased nuclear localisation of Yap1 upon *Fat4* knock-down or accompanying *Amotl1* forced nuclear translocation (Supplementary figure 3k-n), (2) the increased nuclear localisation of *Amotl1* upon *Fat4* knock-down (Supplementary figure 3o).

9) Fig4h. Overexpressed Amot (with 3 nuclear tags) has a 30% increased Ki67, in turn "rescued" by YAP inactivation. Why are this assumed to be serial, rather than parallel regulations? There is no way to distinguish this from what shown here.

It is a combination of observations which made us conclude on a serial model. Yap1 and Amotl1 are known to be strong partners. Forcing Amotl1 to localise in the nucleus increases Yap1 nuclear localisation, as we now further quantify by fractionated Western Blots (Supplementary figure 3m-n). In reverse, interfering with Yap1 expression, prevents nuclear Amotl1 to activate cardiomyocyte proliferation. These data support a serial regulation, although we do not rule out the influence of other factors on the regulation of Yap1 and Amotl1.

As stated by Reviewer1 "in the absence of any other data or understanding of the nature of the players in this pathway, it would only be possible to take the neutral view that one pathway modulates the effects of the other. It does not establish epistasis. All networks are complex with multiple cross-regulatory interactions and feedbacks and parallelism or serialism are not established in any epistasis assay. This data adds to the model in a compelling way. If Amotl1 becomes more compellingly localised to the nucleus in Fat4 mutants, in which cardiomyocyte proliferation is stimulated, then the fact that a nuclear localising form of Amotl1 stimulated proliferation supports the "serial" model in the absence of evidence to suggest a parallel model."

10) As previously requested, the interaction with overexpressed protein is suggestive at best, but not acceptable by itself. Endogenous proteins have not been provided during revision. How do we know that those associations are real?

We agree that data on endogenous proteins would be more significant than data on overexpressed proteins. However, investigation of the interaction between endogenous proteins is limited by the available tools to detect them (antibodies). It is a common practice to use tagged proteins in this case.

11) What is the rationale of jumping from an siRNA data (and see point 13) to overexpressing a nuclear AMOTL1?

It is a common practice to test the effect of the loss-of-function and the gain-of-function. We have initially tested overexpression of Amotl1 (without the nuclear signals). In this case, Amotl1 is sequestered in vesicles in the cytoplasm and thus cannot exert its role in the nucleus, as observed in Fat4 mutants. This is why we finally overexpressed nuclear Amotl1.

12) What are the consequences of nuclear AMOT for YAP, since YAP does not change its nuclear localization after nuclear-Amot overexpression (Fig 4d),

Yap1 does change its nuclear localisation (see the bright spots of Yap1 in the full resolution image of Fig. 4d, co-localising with its partner nlsAmotl1). We have now further quantified this by fractionated Western Blots (Supplementary figure 3m-n).

13) Fig 4f: how can lane 2 and 3 be statistically significant?

This is the result of a rigorous ANOVA test.

14) Fig 4f: the interpretation is really biased. I may say, by comparing lane 3 and 4, that it is actually fat4 siRNA that rescues Amot siRNA. So, I may conclude that Fat4 is downstream of Amotl1, and not viceversa. There is no redundant evidence (let alone YAP specificity) supporting the view that what we are looking at is the model they are portraying.

As stated by Reviewer1 "They show that the impact of Fat4 knockdown on relative percent of Ki67+ cells is ameliorated by knockdown of Amotl1, as expected if Amotl1 is downstream of Fat4 in the way envisaged, AND that the strong effects of Amotl1 knockdown alone are dependent on the presence of Fat4. Moreover, looking at the broader context, Fat4 is a membrane adhesion protein while Amotl1 is a multifunctional protein with roles at the membrane and in the nucleus and for which there is evidence (albeit in vitro) for an interaction with both Fat4 and Yap1". We have now clarified the text accordingly

In sum, even if we do sidestep the request of Rev.3 to show AMOTL1 genetic requirement (that I would consider mandatory at this stage in light of the complexity of the various AMOT functions so far reported), the paper is anyway not showing any rescue with YAP conditional knockouts (there is actually a negative data, see point 6 above). We are left with a fat 4 whole embryonic phenotype, and siRNA data showing in vitro results that are quantitatively minor and open to different interpretations.

There is no validated connection between the Fat4 phenotype shown in Fig1 and YAP; or between FAT and

AMOTL1 (and these mice are available so their refusal to carry out these crosses is difficult to understand) or between AMOT11 (of unknown relevance in the heart) and YAP.

The problem of the MS is not on mechanisms but on the soundness of the data and gene-interactions provided (still at the functional level).

The reviewer ignores the data we have presented in the manuscript and the technical challenges of some questions.

We agree with the reviewer that in vivo data on Amotl1 will be important. Amotl1 knock-out mice have indeed been generated but die at implantation stages due to defects in the placenta (personal communication). As shown in the second run of our revision, we have attempted to knock-down Amotl1 in cardiomyocytes in vivo by infection with AAV9 viruses. However, we have faced technical limitations when 6 different shRNA vectors have not been efficient in knocking-down Amotl1.

The final drawing goes well beyond the data.

In their response to the reviewers:

"In the heart we show that Amotl1 has an inhibitory role in the presence of Fat4, by sequestering Yap1 out of the nucleus"

this is shown nowhere in the paper. There is an siRNA for AMOT11 opposing basal ki67 in cultured cardiomyocytes.

We show in Fat4 mutants and Fat4 knocked-down cardiomyocytes that activated cardiomyocyte proliferation is associated with more nuclear Amotl1 and Yap1. We also provide evidence for a co-localisation of Fat4, Amotl1 and Yap1. And we show that forcing Amotl1 to localise in the nucleus increases Yap1 nuclear localisation and that interfering with Yap1 expression, prevents nuclear Amotl1 to activate cardiomyocyte proliferation.

REVIEWERS' COMMENTS:

Reviewer #1 (Remarks to the Author):

With respect to the response to point 1 of Reviewer 2 concerning the difference in ventricular area between germ line Fat4 KO (1.7) and conditional Mesp1-Cre KO (1.2), the response is reasonable. Conditional KOs are rarely perfect and the authors have now quantified the number of CMs showing deletion of an mTmG CRE reporter using the CRE strategy - this is 70%. This has to be considered an indicative figure. The actual deletion at the Fat4 locus may be different. However, given this finding, a change of ventricular area of 1.2 instead of 1.7 is not so unexpected. The point comes back to whether the data is statistically robust, which it appears to be. Responses to other points including HW/TL or HW/BW are reasonable. HW/BW and HW/TL are indeed gold standards for revealing growth perturbations, but are used almost exclusively in the adult. Neonates are a different matter, with body and heart size influenced by competition for nutrients between siblings on the teat. This is highly dependent on litter size. Hence variance can be expected to be higher. The nuclear/cytoplasmic fractionations with Western blotting are a strong addition.

The responses to the issues raised by Reviewer 3 are reasonable and draw on the new data that accompanies this revision. I might highlight the following:

1. Point 4. The morphological comparison between Mesp1-Cre KOs at fetal stages as shown in one of the Suppl Figs, does not show differences in ventricular wall thickness but proliferation indexes are significant at the later time points. It is perfectly reasonable to think that ventricular area differences might take time to develop especially since a component of the phenotype is driven by CM hypertrophy.
2. Point 6. The change in ventricular area in Fat4 KOs in which one copy of the Yap gene has been deleted does not quite reach significance ($p=0.068$). It is reasonable to declare this as a trend in light of the significant changes in pH3+ cells and proportion of CMs. This finding must be seen in the context of deletion of a SINGLE allele of Yap where the morphological phenotype might be expected to be mild.
3. Point 10 on use of tagged proteins. This is a reasonable response - their use is very widespread and acceptable in the absence of appropriate high quality antibodies. One must of course be cautious about over-expression, and results should be taken not in isolation but as part of a more complete series of investigations. I think their use here is reasonable and interpretations are contextual.
4. Point 11. There is considerable power in the use of gain-of-function and loss-of-function in dissecting mechanism.

--

Reviewer #2 (Remarks to the Author):

The authors have addressed my prior comments and I am enthusiastic for this manuscript to move forward.

I have two comments on the revisions:

1. I recommend that this comment be modified or removed: "Given that the ventricles at birth contain about 1 million cardiomyocytes, an increased percentage of cardiomyocytes by 5% would represent about 50,000 new cardiomyocytes." . 5% is the change in the fraction of cardiomyocytes, which depends on both the number myocytes and non-myocytes. This number is sensitive to the method used for dissociation; the 80% CM fraction is quite high, compared to an estimate of about 30% CM from adult heart (see Tallquist and colleagues, Circ Res, 2016). A calculation of the number of additional CMs based on this number is probably putting too much faith in the percentage.
2. In the cell fractionation studies shown in the supplement, what is pH3? Is this phosphohistone H3? This is an odd choice for internal control marker, especially since the amount of phosphohistone H3 would be expected to change by intervention and therefore not be suitable for use as an internal control. Total histone H3 would be more appropriate.

Response to reviewers

Reviewer #2

The authors have addressed my prior comments and I am enthusiastic for this manuscript to move forward. I have two comments on the revisions:

1. I recommend that this comment be modified or removed: "Given that the ventricles at birth contain about 1 million cardiomyocytes, an increased percentage of cardiomyocytes by 5% would represent about 50,000 new cardiomyocytes." 5% is the change in the fraction of cardiomyocytes, which depends on both the number of myocytes and non-myocytes. This number is sensitive to the method used for dissociation; the 80% CM fraction is quite high, compared to an estimate of about 30% CM from adult heart (see Tallquist and colleagues, Circ Res, 2016). A calculation of the number of additional CMs based on this number is probably putting too much faith in the percentage.

We have removed this sentence

2. In the cell fractionation studies shown in the supplement, what is pH3? Is this phosphohistone H3? This is an odd choice for internal control marker, especially since the amount of phosphohistone H3 would be expected to change by intervention and therefore not be suitable for use as an internal control. Total histone H3 would be more appropriate.

PH3 is indeed phospho-histone H3. We explain in the Methods section - Immunoprecipitation and Western blots "The nuclear marker Phospho-histone H3 and the cytoplasmic marker Gapdh were used as controls of the fractionation". We agree that it would have been a problem to use PH3 if it had been used as a reference level for quantification. However, PH3 was not used in our quantification of the ratio nuclear/cytoplasmic Yap1. It is only used as a nuclear marker to control the cellular fractionation.